# SAVA: SCALABLE LEARNING-AGNOSTIC DATA VALUATION

**Samuel Kessler**[*][†]
Microsoft
`samuel.kessler@microsoft.com`

**Tam Le**[*]
The Institute of Statistical Mathematics / RIKEN AIP
`tam@ism.ac.jp`

**Vu Nguyen**
Amazon
`vutngn@amazon.com`

## ABSTRACT

Selecting data for training machine learning models is crucial since large, web-scraped, real datasets contain noisy artifacts that affect the quality and relevance of individual data points. These noisy artifacts will impact model performance. We formulate this problem as a data valuation task, assigning a value to data points in the training set according to how similar or dissimilar they are to a clean and curated validation set. Recently, *LAVA* (Just et al., 2023) demonstrated the use of optimal transport (OT) between a large noisy training dataset and a clean validation set, to value training data efficiently, without the dependency on model performance. However, the *LAVA* algorithm requires the entire dataset as an input, this limits its application to larger datasets. Inspired by the scalability of stochastic (gradient) approaches which carry out computations on *batches* of data points instead of the entire dataset, we analogously propose *SAVA*, a scalable variant of *LAVA* with its computation on batches of data points. Intuitively, *SAVA* follows the same scheme as *LAVA* which leverages the hierarchically defined OT for data valuation. However, while *LAVA* processes the whole dataset, *SAVA* divides the dataset into batches of data points, and carries out the OT problem computation on those batches. Moreover, our theoretical derivations on the trade-off of using entropic regularization for OT problems include refinements of prior work. We perform extensive experiments, to demonstrate that *SAVA* can scale to large datasets with millions of data points and does not trade off data valuation performance. Our code is available at `https://github.com/skezle/sava`.

## 1 INTRODUCTION

Neural scaling laws empirically show that the generalization error decreases according to a power law as the data a model trains on increases. This has been shown for natural language processing, vision, and speech (Kaplan et al., 2020; Henighan et al., 2020; Rosenfeld et al., 2019; Zhai et al., 2022; Radford et al., 2023). However, training neural networks on larger and larger datasets for moderate improvements in model accuracy is inefficient. Furthermore, production neural network models need to be continuously updated given new utterances that enter into common everyday parlance (Lazaridou et al., 2021; Baby et al., 2022). It has been shown both in theory and in practice that sub-power law, exponential scaling of model performance with dataset size is possible by carefully selecting informative data and pruning uninformative data (Sorscher et al., 2022), and that generalization improves with training speed (Lyle et al., 2020). Therefore, valuing and selecting data

---

[*]co-first authors [†]work done while at the University of Oxford.

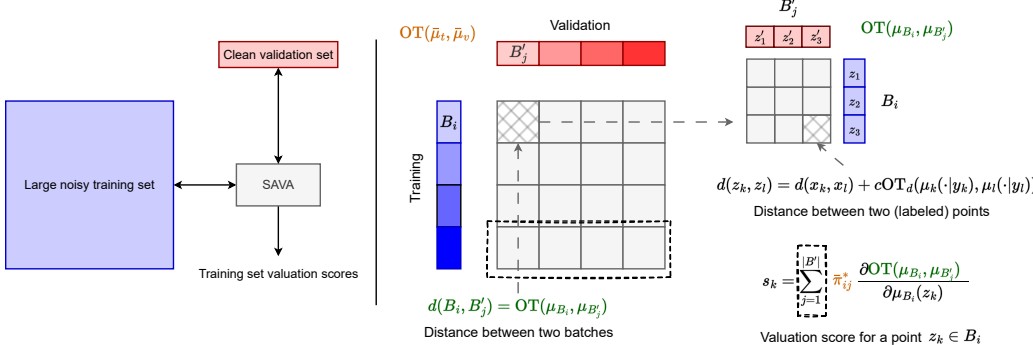

Figure 1: **Overview of the proposed *SAVA* method**. On the left-hand side, *SAVA* values data points in a noisy training dataset by comparing to a clean validation dataset. *SAVA* performs scalable data valuation by solving multiple cheap and small OT problems on *batches* of data points (on the right-hand side). Notations in Orange denote OT distances and plans over training and validation batches, while notations in Green denote OT distances over data points in a batch. $\mathrm{OT}(\bar{\mu}_t, \bar{\mu}_v)$ denotes the OT distance between training and validation batches and $\bar{\pi}^*(\bar{\mu}_t, \bar{\mu}_v)$ is the associated OT plan in Eq. (11). $\mathrm{OT}(\mu_{B_i}, \mu_{B'_j})$ is the OT distance between the batch $B_i$ from the training set and the batch $B'_j$ in the validation set where we use the feature-label distance in Eq. (1) as the ground cost for labeled data points in these batches. $s_k$ is *SAVA*'s final valuation score for training labeled data point $z_k$ in Eq. (12). The hatched box denotes the summation over the validation batches to value the data point $z_k$. We provide a visualization of these artifacts generated by *SAVA* in Figure 12.

points that are informative: which have not been seen by the model, which do not have noisy labels, and which are relevant to the task we want to solve—are not outliers—can help to not only decrease training times, and reduce compute costs, but also improve overall test performance (Mindermann et al., 2022; Tirumala et al., 2023).

Popular data selection and data pruning methods use variations of the model loss to value data points (Jiang et al., 2019; Pruthi et al., 2020; Paul et al., 2021). Crucially, these methods depend on the model used, and they are vulnerable to prioritizing data points with noisy labels or noisy features; data points that do not resemble the target validation set. When treating data valuation as a function of model performance, we introduce a dependency on a neural network model. This is the case when valuing a point using the leave-one-out (LOO) error, i.e., the change of model performance when the point is omitted from training. To rid our dependence on a neural network model, a promising idea is to leverage optimal transport (OT) between a training distribution and a clean validation distribution as a proxy to directly measure the value of data points in the training set in a model-agnostic fashion. In particular, the validation performance of each point in the training set can be estimated using the hierarchically defined Wasserstein between the training and the validation set (Alvarez-Melis & Fusi, 2020; Just et al., 2023). *LAVA* (Just et al., 2023) has been shown to successfully value data by measuring the sensitivity of the hierarchically defined Wasserstein between training data points and validation distributions in a model-agnostic fashion. However, *LAVA* requires significant RAM consumption since its memory complexity grows quadratically $\mathcal{O}(N^2)$ with the dataset size $N$. This hinders *LAVA* from scaling to large datasets.

In this paper, we present *SAVA*, a scalable variant of *LAVA*, for data valuation. Our method completely addresses the bottleneck of RAM requirements in *LAVA*. Intuitively, *SAVA* performs the OT computations on batches instead of on the entire dataset like *LAVA* (hence the analogy to the stochastic approach to (sub)gradient computation). Specifically, *SAVA* uses ideas from hierarchical optimal transport (Yurochkin et al., 2019; Lee et al., 2019) to enable OT calculations on batches instead of the entire dataset. We can scale up OT-based data valuation using *SAVA* to large real-world web-scrapped datasets. On benchmark problems *SAVA* performs comparably to *LAVA* while being able to scale to datasets two orders of magnitude larger without memory issues, while *LAVA* is limited due to hardware memory constraints.

**Contributions:** In summary, our contributions are three-fold as follows:

- We introduce a novel scalable data valuation method called *SAVA* that leverages the (sub)gradient of hierarchical optimal transport which performs OT computations on batches of data points, enabling OT-based data valuation to large datasets.

- We correct the essential theoretical result on the trade-off for using entropic regularization to compute the OT (sub)gradient in *LAVA* (Just et al., 2023, Theorem 2).[1] Consequently, by building upon our refined theory, we derive the exact trade-off to estimate the (sub)gradient of hierarchical OT with entropic regularization in our framework.

- We provide an extensive experimental analysis to demonstrate the improved scalability with increasing dataset sizes with respect to baselines.

## 2 OPTIMAL TRANSPORT FOR DATA VALUATION

### 2.1 OPTIMAL TRANSPORT FOR LABELED DATASETS

Let $\mathcal{X}$ be the feature space, and $V$ be the number of labels. We write $f_t : \mathcal{X} \mapsto \{0,1\}^V$ and $f_v : \mathcal{X} \mapsto \{0,1\}^V$ for the labeling functions for training and validation data respectively. Given the training set $\mathbb{D}_t = \{(x_i, f_t(x_i))\}_{i=1}^N$ and the validation set $\mathbb{D}_v = \{(x_i', f_v(x_i'))\}_{i=1}^{N'}$, the corresponding measures for sets $\mathbb{D}_t, \mathbb{D}_v$ are $\mu_t(x,y) = \frac{1}{N} \sum_{i=1}^N \delta_{(x_i,y_i)}$ and $\mu_v(x',y') = \frac{1}{N'} \sum_{i=1}^{N'} \delta_{(x_i',y_i')}$ respectively where $\delta$ is the Dirac function, and $y, y'$ are labels of $x, x'$ respectively. For simplicity, let $\mathcal{Z} = (\mathcal{X}, \mathcal{Y})$ where $\mathcal{Y}$ is the space of labels. For ease of reading, we summarize the notations in Table 1.

Following Alvarez-Melis & Fusi (2020), we compute the distance between two labels by leveraging the OT distance between the conditional distributions of the features given each label, i.e., $\mu_t(\cdot|y_t) = \frac{\mu_t(\cdot)I[f_t(\cdot)=y_t]}{\int \mu_t(\cdot)I[f_t(\cdot)=y_t]}$ for label $y_t$ in $\mu_t$, where $I$ is the indicator function.

Let d be the metric of the feature space $\mathcal{X}$, e.g., the Euclidean distance. The distance between labels $y_t$ and $y_v$ is $\mathrm{OT}_d(\mu_t(\cdot|y_t), \mu_v(\cdot|y_v))$, i.e., the metric of the label space $\mathcal{Y}$. Consequently, the cost between feature-label pairs in $\mathcal{Z} = (\mathcal{X}, \mathcal{Y})$ is

$$\mathrm{C}((x_t, y_t), (x_v, y_v)) = \mathrm{d}(x_t, x_v) + c\,\mathrm{OT}_d(\mu_t(\cdot|y_t), \mu_v(\cdot|y_v)), \tag{1}$$

where $c > 0$ is a weight coefficient. Therefore, with the cost matrix C, we can use the OT on the represented measures $\mu_t, \mu_v$ to compute the distance between the training and validation sets, i.e., $d(\mathbb{D}_t, \mathbb{D}_v)$, without relying on external models or parameters as follows

$$d(\mathbb{D}_t, \mathbb{D}_v) := \mathrm{OT}_{\mathrm{C}}(\mu_t, \mu_v) = \min_{\pi \in \Pi(\mu_t, \mu_v)} \int_{\mathcal{Z} \times \mathcal{Z}} \mathrm{C}(z, z') d\pi(z, z'), \tag{2}$$

where $\Pi(\mu_t, \mu_v)$ is the set of transportation couplings with marginals as $\mu_t$ and $\mu_v$. To simplify notations, we drop C, and use OT when the context is clear. We further write $\pi^*$ for the optimal transport plan in Eq. (2). In practice, we leverage the entropic regularization (Cuturi, 2013) to reduce the OT complexity into quadratic (from super cubic) w.r.t. the number of input supports, defined as

$$\mathrm{OT}_\varepsilon(\mu_t, \mu_v) = \min_{\pi \in \Pi(\mu_t, \mu_v)} \int_{\mathcal{Z} \times \mathcal{Z}} \mathrm{C}(z, z') d\pi(z, z') + \varepsilon H(\pi \mid \mu_t \otimes \mu_v), \tag{3}$$

where $\otimes$ is the product measure operator, and $H(\pi \mid \mu_t \otimes \mu_v) = \int_{\mathcal{Z} \times \mathcal{Z}} \log\left(\frac{d\pi}{d\mu_t d\mu_v}\right) d\pi$.

Additionally, the OT problem in Eq. (2) is a constrained convex minimization, it is naturally paired with a dual problem, i.e., constrained concave maximization problem, as follows:

$$\mathrm{OT}_{\mathrm{C}}(\mu_t, \mu_v) = \max_{(f,g) \in \mathcal{R}(\mathrm{C})} \langle f, \mu_t \rangle + \langle g, \mu_v \rangle, \tag{4}$$

where $\mathcal{R}(\mathrm{C}) = \{(f,g) \in \mathcal{C}(\mathcal{Z}) \times \mathcal{C}(\mathcal{Z}) : \forall(z, z'), f(z) + g(z') \leq \mathrm{C}(z, z')\}$, $\mathcal{C}$ is a collection of continuous functions, $\langle f, \mu_t \rangle = \int_{\mathcal{Z}} f(z) d\mu_t(z)$, and similarly $\langle g, \mu_v \rangle = \int_{\mathcal{Z}} g(z) d\mu_v(z)$.

### 2.2 LAVA: DATA VALUATION VIA CALIBRATED OT GRADIENTS

As discussed in Just et al. (2023), the OT distance is known to be insensitive to small differences while also being not robust to large deviations. This feature is naturally suitable for detecting abnormal

---

[1]See Appendix C.1 for the detailed discussion.

Table 1: A summary of notations.

| Notation | Definition |
|---|---|
| $z = (x, y) \in \mathbb{R}^{|\mathcal{X}|} \times \{0,1\}^V$ | Data point feature and label where $V$ is #label and $\mathcal{X}$ is a feature space |
| $\mathbb{D}_t, \mathbb{D}_v$ | Datasets for training $\mathbb{D}_t = \{(x_i, f_t(x_i))\}_{i=1}^N$ and validation $\mathbb{D}_v = \{(x'_j, f_v(x'_j))\}_{j=1}^{N'}$ |
| $B = \{B_i\}_{i=1}^{K_t}, B' = \{B'_j\}_{j=1}^{K_v}$ | Disjoined batches for $\mathbb{D}_t$ and $\mathbb{D}_v$ where $K_t, K_v$ are the number of batches |
| $B_i = \{z_k\}_{k=1}^{N_i}$ | Batch of data points where $N_i$ is the size of the training batch $B_i$ |
| $B'_j = \{z_l\}_{l=1}^{N'_j}$ | Batch of data points where $N'_j$ is the size of the validation batch $B'_j$ |
| $\mu_{B_i} = \frac{1}{N_i} \sum_{t=1}^{N_i} \delta_{(z_t)}$ | Measure over labeled data points for the batch $B_i$ |
| $\mu_t(x,y) = \frac{1}{N} \sum_{i=1}^N \delta_{(x_i, y_i)}$ | Measure over training set |
| $\mu_v(x,y) = \frac{1}{N'} \sum_{i=1}^{N'} \delta_{(x_i, y_i)}$ | Measure over validation set |
| $\bar{\mu}_t = \frac{1}{K_t} \sum_{i=1}^{K_t} \delta_{(B_i)}$ | Measure over batches for the training set |
| $\bar{\mu}_v = \frac{1}{K_v} \sum_{j=1}^{K_v} \delta_{(B'_j)}$ | Measure over batches for the validation set |
| $\bar{C} \in \mathbb{R}_+^{K_t \times K_v}$ | Cost matrix over *batches*, each element $\bar{C}_{i,j} = d(B_i, B'_j)$ |
| $C \in \mathbb{R}_+^{N_i \times N'_j}$ | Cost matrix over *labeled data points* within $B_i$ and $B'_j$, each element $C_{kl} = d(z_k, z'_l)$ |
| $f^* \in \mathbb{R}^{N_i}, g^* \in \mathbb{R}^{N'_j}$ | Dual solutions of the OT over a cost matrix $C \in \mathbb{R}^{N_i \times N'_j}$ |
| $\pi^* \in \mathbb{R}^{N_i \times N'_j}$ | OT plan over a cost matrix $C \in \mathbb{R}^{N_i \times N'_j}$ |
| $\bar{f}^* \in \mathbb{R}^{K_t}, \bar{g}^* \in \mathbb{R}^{K_v}$ | OT dual solutions over a cost matrix over batches $\bar{C} \in \mathbb{R}^{K_t \times K_v}$ |
| $\bar{\pi}^* \in \mathbb{R}^{K_t \times K_v}$ | OT plan over a cost matrix between batches $\bar{C} \in \mathbb{R}^{K_t \times K_v}$ |
| $\mathrm{OT}_C(\mu_t, \mu_v)$ | OT solution to the opt. problem with cost $C$ over training and validation set measures |

data points, i.e., disregarding normal variations in distances between clean data while being sensitive to abnormal distances of outlying points. Therefore, the (sub)gradient of the OT distance w.r.t. the probability mass associated with each point can be leveraged as a surrogate to measure the contribution of that point. More precisely, the (sub)gradient of the OT distance w.r.t. the probability mass of data points in the two datasets can be expressed:

$$\nabla_{\mu_t} \mathrm{OT}(\mu_t, \mu_v) = (f^*)^T, \ \nabla_{\mu_v} \mathrm{OT}(\mu_t, \mu_v) = (g^*)^T. \tag{5}$$

The dual solution is unique up to a constant due to the redundant constraint $\sum_{i=1}^N \mu_t(z_i) = \sum_{i=1}^M \mu_v(z'_i) = 1$. Therefore, for measuring the subgradients of the OT w.r.t. the probability mass of a given data point in each dataset, Just et al. (2023) propose to calculate the *calibrated gradients* (i.e., a sum of all elements equals to zero)[2] as

$$\frac{\partial \mathrm{OT}(\mu_t, \mu_v)}{\partial \mu_t(z_i)} = f_i^* - \sum_{j \in \{1,\dots,N\} \setminus i} \frac{f_j^*}{N-1}. \tag{6}$$

The calibrated gradients predict how the OT distance changes as more probability mass is shifted to a given data point. This can be interpreted as a measurement of the contribution of the data point to the OT. Additionally, if we want a training set to match the distribution of the validation dataset, then either removing or reducing the mass of data points with large positive gradients, while increasing the mass of data points with large negative gradients can be expected to reduce their OT distance. Therefore, as demonstrated in Just et al. (2023), the calibrated gradients provide a powerful tool to detect and prune abnormal or irrelevant data in various applications.

**Memory limitation.** While being used with the current best practice, the Sinkhorn algorithm for entropic regularized OT (Cuturi, 2013) still runs in quadratic memory complexity $\mathcal{O}(N^2)$ with the dataset size $N$, as it requires performing operations on the entire dataset, using the full pairwise cost matrix. Consequently, the memory and RAM requirements for the Sinkhorn algorithm primarily depend on the dataset size $N$. Additionally, notice that a dense square (float) matrix of size $N = 10^5$ will require at least 74 GB of RAM and $N = 10^6$ will take 7450 GB of RAM which is prohibitively expensive. Therefore, *LAVA* is limited to small datasets.

**Scalability.** Inspired by the scalability of stochastic gradient approaches where the computation is carried out on batches of data points instead of the whole dataset as in the traditional gradient, we follow this simple but effective scheme to propose an analog for OT, named *SAVA* which is a scalable variant of *LAVA* with its OT computation on batches. Intuitively, *SAVA* also leverages the

---

[2]To remove the degree of freedom which comes from the fact that one among all row/column sum constraints for the transport polytope is redundant, among the OT subgradients, we fix the zero-sum subgradient following the convention in Cuturi & Doucet (2014) and the calibrated approach in *LAVA* (Just et al., 2023).

---

**Algorithm 1** Scalable Data Valuation (*SAVA*) algorithm. More concretely, in Lines 1–5, we solve multiple OT problems between batches. In Line 6, we solve the OT problem across batches: $\mathrm{OT}_{\bar{\mathrm{C}}}(\bar{\mu}_t, \bar{\mu}_v)$, to obtain $\bar{\pi}^*(\bar{\mu}_t, \bar{\mu}_v)$. In Lines 7–10, we estimate valuation scores for training data using the plan $\bar{\pi}^*(\bar{\mu}_t, \bar{\mu}_v)$ and potentials $f^*(\mu_{B_i}, \mu_{B'_j})$ computed in the previous steps.

---

**Input:** a threshold $\varepsilon$ for Sinkhorn algorithm, let $z = (x, y)$
**Output:** training data values $s_k$ for all $k \in [N_i]$ for all $i \in [K_t]$.
1 **for** $i = 1, ..., K_t$ **do**
2     **for** $j = 1, ..., K_v$ **do**
3        Compute $\mathrm{C}_{kl}(B_i, B'_j), \forall k \in [N_i], \forall l \in [N'_j]$ by using Eq. (8).
4        Compute $f^*(\mu_{B_i}, \mu_{B'_j}), g^*(\mu_{B_i}, \mu_{B'_j})$ by solving $\mathrm{OT}_{\mathrm{C}}(\mu_{B_i}, \mu_{B'_j})$.
5        Set $\bar{\mathrm{C}}_{ij}(\bar{\mu}_t, \bar{\mu}_v) = \mathrm{OT}_{\mathrm{C}}(\mu_{B_i}, \mu_{B'_j})$.          // distance $d(B_i, B'_j)$ on batches.
6 Compute $\bar{\pi}^*(\bar{\mu}_t, \bar{\mu}_v) \in \mathbb{R}^{K_t \times K_v}$ by solving $\mathrm{OT}_{\bar{\mathrm{C}}}(\bar{\mu}_t, \bar{\mu}_v)$ using Eq. (11).
7 **for** $i = 1, ..., K_t$ **do**
8     **for** $k = 1, ..., N_i$ **do**
9        Compute $\frac{\partial \mathrm{OT}(\mu_{B_i}, \mu_{B'_j})}{\partial \mu_{B_i}(z_k)}, \ \forall j \in [K_v]$ using Eq. (14).
10        Compute $s_k = \frac{\partial \mathrm{HOT}(\mu_t, \mu_v)}{\partial \mu_t(z_l)}$ using Eq. (13).      // valuation score for $z_k \in B_i$.

---

hierarchically defined OT as in *LAVA*, but it performs OT on batches of data points instead of on the entire dataset as in *LAVA*.

Notice that the scalable OT-based data valuation (i.e., *SAVA*) we will introduce in the next section focuses on the *(sub)gradient* of the OT instead of the OT *distance* itself. Therefore, several popular scalable OT approaches such as sliced-Wasserstein (Rabin et al., 2011; Bonneel et al., 2015; Nguyen et al., 2024), tree-sliced-Wasserstein (Le et al., 2019; 2024b; Tran et al., 2025a;b), or Sobolev transport (Le et al., 2022; 2023; 2024a), may not be suitable since they leverage local structures on supports (e.g., line, tree, or graph structure respectively) to scale up the computation of OT distance. In the next section, we introduce our novel scalable approach for computing the (sub)gradient of the OT using hierarchical OT (Yurochkin et al., 2019; Lee et al., 2019). We focus on the problem of data valuation but our work can be applied to other large dataset applications where the (sub)gradient of the OT is required.

## 3 SAVA: SCALABLE DATA VALUATION

In this section, we present *SAVA*, a scalable data valuation method, scaling *LAVA* to large-scale datasets. Instead of solving a single, but expensive OT problem for distributions on the entire datasets, i.e., $\mathrm{OT}(\mu_t, \mu_v)$ in *LAVA* with the pairwise cost matrix size $\mathbb{R}^{N \times N'}$, we consider solving multiple cheaper OT problems for distributions on batches of data points. For this purpose, our algorithm performs data valuation on two levels of hierarchy: across batches, and across data points within two batches. Thus, *SAVA* can complement *LAVA* for large-scale applications.

**Hierarchical OT.** We follow the idea in hierarchical OT approach (Yurochkin et al., 2019; Lee et al., 2019) to partition the training dataset $\mathbb{D}_t$ of $N$ samples into $K_t$ disjoint batches $B = \{B_i\}_{i=1}^{K_t}$. Similarly, for the validation set $\mathbb{D}_v$ of $N'$ samples into $K_v$ disjoint batches $B' = \{B'_j\}_{j=1}^{K_v}$. Additionally, for all $i \in [K_t], j \in [K_v]$, let the number of samples in batches $B_i, B'_j$ as $N_i, N'_j$ respectively. The corresponding measures of the training and validation sets w.r.t. the batches are defined as: $\bar{\mu}_t(B) = \frac{1}{K_t} \sum_{i=1}^{K_t} \delta_{(B_i)}$ and $\bar{\mu}_v(B') = \frac{1}{K_v} \sum_{j=1}^{K_v} \delta_{(B'_j)}$ respectively. We then define a distance between the datasets as the hierarchical optimal transport (HOT) between the measures $\mu_t, \mu_v$ as OT distance for corresponding represented measures on batches, i.e., $\bar{\mu}_t$ and $\bar{\mu}_v$ as in §2.1 as follows:

$$d(\mu_t, \mu_v) := \mathrm{HOT}(\mu_t, \mu_v) := \mathrm{OT}(\bar{\mu}_t, \bar{\mu}_v). \tag{7}$$

It is worth noting that HOT finds the optimal coupling at the batch level, but not at the support data point level as in the classic OT. Therefore, it can be seen that

$$\mathrm{OT}(\mu_t, \mu_v) \le \mathrm{HOT}(\mu_t, \mu_v),$$

where the equality trivially happens when either each batch only has one support or each dataset only has one batch.

Our goal is to estimate the (sub)gradient $\frac{\partial d(\mu_t, \mu_v)}{\partial \mu_t(z_k)} = \frac{\partial \text{HOT}(\mu_t, \mu_v)}{\partial \mu_t(z_k)}$ where $\text{HOT}(\mu_t, \mu_v)$ is defined in the Eq. (7). Computing this derivative involves two OT estimation steps including (i) OT between individual data points within two batches to compute $d(B_i, B'_j) := \text{OT}(\mu_{B_i}, \mu_{B'_j})$, where $\mu_{B_i}, \mu_{B'_j}$ are corresponding measures for batches $B_i, B'_j$ respectively, and subsequently a cost matrix $\bar{\text{C}}$ on pairwise batches for input measures over batches; (ii) $d(\mu_t, \mu_v) = \text{HOT}(\mu_t, \mu_v) := \text{OT}_{\bar{\text{C}}}(\bar{\mu}_t, \bar{\mu}_v)$.

**Pairwise cost between batches.** We estimate the distance between two batches as the OT problem between $B_i$ and $B_j$, i.e., $d(B_i, B'_j) := \text{OT}(B_i, B'_j)$ as discussed in §2.1 by viewing the OT problem between two labeled (sub)datasets, i.e., batches $B_i$ and $B'_j$.

More precisely, to solve this OT problem, we calculate the pairwise cost for data points between two batches $\text{C}_{kl}(B_i, B'_j) \in \mathbb{R}^{N_i \times N'_j}$, where $\forall k \in [N_i], \forall l \in [N'_j]$, the element $\text{C}_{kl}(B_i, B'_j)$ is the cost between two labeled data points $(x_k, y_k) \in B_i$ and $(x'_l, y'_l) \in B'_j$, calculated as

$$\text{C}_{kl}(B_i, B'_j) = \text{d}(x_k, x'_l) + c\, \text{OT}_{\text{d}}(\mu_{B_i}(\cdot | y_k), \mu_{B'_j}(\cdot | y'_l)). \tag{8}$$

Given the cost matrix $\text{C}(B_i, B'_j)$, we solve $\text{OT}_{\text{C}}(\mu_{B_i}, \mu_{B'_j})$ to get dual solutions $f^*(\mu_{B_i}, \mu_{B'_j}), g^*(\mu_{B_i}, \mu_{B'_j})$, and the OT distance for $d(B_i, B'_j) = \text{OT}_{\text{C}}(\mu_{B_i}, \mu_{B'_j})$.

We repeat this process and solve the OT problem for each pair $(B_i, B'_j)$, i.e., the OT problem for distributions on batches of data points, across the training and validation datasets, for all $i \in [K_t], j \in [K_v]$. This enables us to define the cost matrix for pairwise batches in $\bar{\mu}_t, \bar{\mu}_v$, denoted as $\bar{\text{C}}(\bar{\mu}_t, \bar{\mu}_v) \in \mathbb{R}_+^{K_t \times K_v}$ where we recall that $K_t$ and $K_v$ are the number of batches in training and validation sets. Hence, for this cost matrix $\bar{\text{C}}$, the element $\bar{\text{C}}_{ij}$ is computed as $\text{OT}(\mu_{B_i}, \mu_{B'_j})$, for all $i \in [K_t], \forall j \in [K_v]$.

**Batch valuation.** Given the pairwise cost matrix across batches $\bar{\text{C}}$, we compute the data valuation for each batch via the (sub)gradient of the distance $\text{OT}_{\bar{\text{C}}}(\bar{\mu}_t, \bar{\mu}_v)$ w.r.t. the probability mass of *batches* in the two datasets, i.e., $\frac{\partial \text{OT}_{\bar{\text{C}}}(\bar{\mu}_t, \bar{\mu}_v)}{\partial \bar{\mu}_t(B_i)}$. These partial derivatives measure the contribution of the batches to the OT distance, i.e., shifting more probability mass to the batch would result in an increase or decrease of the dataset distance, respectively.

More precisely, let $\bar{f}^*, \bar{g}^*$ be the optimal dual variables of $\text{OT}_{\bar{\text{C}}}(\bar{\mu}_t, \bar{\mu}_v)$, then the data valuation of the batch $B_i$ in the training set $\mathbb{D}_t$ is estimated as follows:

$$\frac{\partial \text{OT}_{\bar{\text{C}}}(\bar{\mu}_t, \bar{\mu}_v)}{\partial \bar{\mu}_t(B_i)} = \bar{f}_i^*. \tag{9}$$

Since the optimal dual variables are only unique up to a constant, we follow Just et al. (2023) to normalize these optimal dual variables such that the sum of all elements is equal to zero:

$$\frac{\partial \text{OT}_{\bar{\text{C}}}(\bar{\mu}_t, \bar{\mu}_v)}{\partial \bar{\mu}_t(B_i)} = \bar{f}_i^* - \sum_{j \in \{1, \ldots, K_t\} \setminus i} \frac{\bar{f}_j^*}{K_t - 1}. \tag{10}$$

**Using batch valuation for data point valuation.** After solving the OT problem over batches, we obtain the OT plan $\bar{\pi}^*(\bar{\mu}_t, \bar{\mu}_v)$ which is used to compute the data valuation over individual data point:

$$\bar{\pi}^*(\bar{\mu}_t, \bar{\mu}_v) = \text{diag}(\bar{u}^*) \exp\left(-\frac{\bar{\text{C}}(\bar{\mu}_t, \bar{\mu}_v)}{\varepsilon}\right) \text{diag}(\bar{v}^*), \tag{11}$$

where $\text{diag}(\cdot)$ is matrix diagonal operator, $(\bar{u}^*, \bar{v}^*)$ is a dual solution from Sinkhorn algorithm. Additionally, we have $\bar{u}^* = \exp(-\frac{1}{2} - \frac{\bar{f}^*}{\varepsilon}), \bar{v}^* = \exp(-\frac{1}{2} - \frac{\bar{g}^*}{\varepsilon})$ following Cuturi (2013, Lemma 2).

For a data point $z \in B_i \subset \mathbb{D}_t$, its data valuation score can be computed as

$$\sum_{j=1}^{K_v} \bar{\pi}_{ij}^*(\bar{\mu}_t, \bar{\mu}_v) \frac{\partial \text{OT}(\mu_{B_i}, \mu_{B'_j})}{\partial \mu_{B_i}(z)}. \tag{12}$$

Using HOT, we can measure the (sub)gradients of the OT distance w.r.t. the probability mass of a given data point in each dataset via the calibrated gradient summarized in the following Lemma 1. We refer to Table 1 for the notations.

**Lemma 1.** *The calibrated gradient for a data point $z_k$ in the batch $B_i$ in $\mathbb{D}_t$ can be computed as:*

$$\frac{\partial \text{HOT}(\mu_t, \mu_v)}{\partial \mu_t(z_k)} = \sum_{j=1}^{K_v} \bar{\pi}_{ij}^*(\bar{\mu}_t, \bar{\mu}_v) \frac{\partial \text{OT}(\mu_{B_i}, \mu_{B_j'})}{\partial \mu_{B_i}(z_k)}, \tag{13}$$

*where the calibrated gradient of OT for measures on batches is calculated as follows:*

$$\frac{\partial \text{OT}(\mu_{B_i}, \mu_{B_j'})}{\partial \mu_{B_i}(z_k)} = f_k^*(\mu_{B_i}, \mu_{B_j'}) \quad - \sum_{l \in \{1,\dots,N_i\} \setminus k} \frac{f_l^*(\mu_{B_i}, \mu_{B_j'})}{N_i - 1}, \forall j \in [K_v]. \tag{14}$$

It is common practice to compute the OT via its entropic regularization, using the Sinkhorn algorithm (Cuturi, 2013). We quantify the deviation in the calibrated gradients caused by the entropy regularizer. This analysis shows the potential impact of the deviation on applications built on these gradients.

We *refine* the theoretical result (Just et al., 2023, Theorem 2), which uses entropic regularization to approximate the OT (sub)gradient in two aspects. Firstly, we take into account the non-negative constraint of the OT plan $\pi \geq 0$ in the dual formulation. Secondly, we correct the discrete formulation for entropic regularization.[3]

**Theorem 2** (Refined Theorem 2 in Just et al. (2023)). *The difference between calibrated gradients for two data points in $\mathbb{D}_t$ is*

$$\frac{\partial \text{OT}(\mu_t, \mu_v)}{\partial \mu_t(z_i)} - \frac{\partial \text{OT}(\mu_t, \mu_v)}{\partial \mu_t(z_k)} = \frac{\partial \text{OT}_\varepsilon(\mu_t, \mu_v)}{\partial \mu_t(z_i)} - \frac{\partial \text{OT}_\varepsilon(\mu_t, \mu_v)}{\partial \mu_t(z_k)}$$
$$+ \varepsilon \frac{N}{N-1} \left( \log \frac{(\pi_\varepsilon^*)_{ij} / \mu_t(z_i)}{(\pi_\varepsilon^*)_{kj} / \mu_t(z_k)} + \frac{h_{kj}^* - h_{ij}^*}{\varepsilon} \right), \tag{15}$$

*where $h^*$ is the corresponding optimal dual variable, accounting for non-negative constraint on the transportation plan $\pi \geq 0$ in the primal OT formulation.*

*Proof.* Refer to Appendix C.2 for the proof. ∎

The term in the brackets (in blue) is the key difference between our new result and the previous one derived in LAVA (Just et al., 2023, Theorem 2).

**Lemma 3** (HOT with entropic regularized OT over batches). *The difference between the calibrated gradients for two data points $\{z_l, z_h\} \in B_i \subset \mathbb{D}_t$ can be calculated as*

$$\frac{\partial \text{HOT}(\mu_t, \mu_v)}{\partial \mu_t(z_k)} - \frac{\partial \text{HOT}(\mu_t, \mu_v)}{\partial \mu_t(z_l)} = \sum_{j=1}^{K_v} \bar{\pi}_{ij}^*(\bar{\mu}_t, \bar{\mu}_v) \left[ \frac{\partial \text{OT}_\varepsilon(\mu_{B_i}, \mu_{B_j'})}{\partial \mu_{B_i}(z_k)} - \frac{\partial \text{OT}_\varepsilon(\mu_{B_i}, \mu_{B_j'})}{\partial \mu_{B_i}(z_l)} \right.$$
$$\left. + \varepsilon \frac{N_i}{N_i - 1} \left( \log \frac{(\bar{\pi}_\varepsilon^*)_{k,j} / \mu_t(z_k)}{(\bar{\pi}_\varepsilon^*)_{l,j} / \mu_t(z_l)} + \frac{h_{lj}^* - h_{kj}^*}{\varepsilon} \right) \right]. \tag{16}$$

*Proof.* Refer to Appendix C.3 for the proof. ∎

We make a similar observation in *LAVA* that the ground-truth (sub)gradient difference between two training points $z_k$ and $z_l$ is calculated based on the HOT formulation and can be approximated by the entropic regularized formulation $\text{OT}_\varepsilon$ over batches, such as via the Sinkhorn algorithm (Cuturi, 2013). In other words, we can calculate the ground-truth difference based on the solutions to the regularized problem plus some calibration terms that scale with $\varepsilon$. In addition, in our case with HOT, the (sub)gradient difference also depends on the additional optimal assignment across batches $\bar{\pi}^*(\bar{\mu}_t, \bar{\mu}_v)$ which is again estimated by the Sinkhorn algorithm.

---

[3]See Appendix C.1 for the detailed discussion.

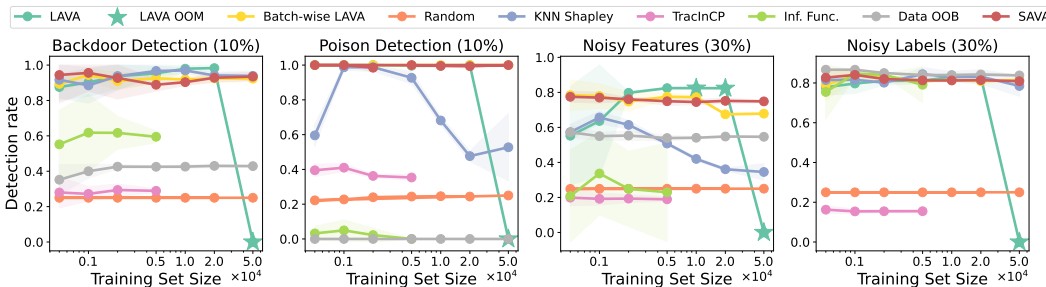

Figure 2: *SAVA* can value the full CIFAR10 dataset with various corruptions, while *LAVA* has out-of-memory (OOM) issues. We sort training examples by the highest OT gradients in Eq. (6) and Eq. (12) for *LAVA* and *SAVA* respectively, and use the fraction of corrupted data recovered for a prefix of size $N/4$ as the detection rate (where $N$ is the training set size). The star symbol (★) denotes the point at which *LAVA* is unable to continue valuing training due GPU out-of memory (OOM) errors.

## 4 PROPERTIES AND DISCUSSIONS

**The *SAVA* algorithm.** We outline the computational steps of *SAVA* in Algorithm 1. In Lines 1–5, we solve multiple OT tasks for data points between two batches $B_i, B'_j$. We obtain the dual solution $f^*(\mu_{B_i}, \mu_{B'_j}) \in \mathbb{R}^{N_i}$. Additionally, these OT distances are used to fill in the cost matrix for pairwise batches $\bar{C}_{ij}(\bar{\mu}_t, \bar{\mu}_v) = \text{OT}_C(\mu_{B_i}, \mu_{B'_j})$. Here, the cost matrix C is of size $N_i \times N'_j$ where the batch sizes $N_i, N'_j \ll N$. The required memory complexity is $\mathcal{O}(N_i \times N'_j)$.

In Line 6 Algorithm 1, we solve the OT problem across batches $\text{OT}_{\bar{C}}(\bar{\mu}_t, \bar{\mu}_v)$ to obtain $\bar{\pi}^*(\bar{\mu}_t, \bar{\mu}_v)$. The cost matrix is small, of size $K_t \times K_v$, so less expensive compared to working with the full dataset Just et al. (2023). Finally, in Lines 7–10, we estimate valuation scores for training data using the auxiliary matrices computed in the previous steps, including $f^*(\mu_{B_i}, \mu_{B'_j})$ and $\bar{\pi}^*(\bar{\mu}_t, \bar{\mu}_v)$.

***SAVA* memory requirements.** The memory complexity of *LAVA* is $\mathcal{O}(N \times N')$ where $N$ and $N'$ are the training and validation dataset sizes. This comes from the main OT step of $\text{OT}(\mu_t, \mu_v)$ over the full cost matrix C of size $N \times N'$ to subsequently calculate the calibrated gradient in Eq. (6). *SAVA* overcomes this limitation by solving multiple smaller OT problems $\text{OT}(\mu_{B_i}, \mu_{B'_j})$ on distributions for batches of data points only: if batches $B_i$ and $B'_j$ are of size $N_i$ and $N'_j$ respectively, then the memory complexity is $\mathcal{O}(N_i \times N'_j)$. See Appendix D for an analysis of the time complexity.

**Practical implementation with caching.** While yielding a significant memory improvement, *SAVA*'s runtime is not necessarily faster than *LAVA*. To speed up *SAVA*, we propose to implement Algorithm 1 by caching the label-to-label costs between points in the validation and training batches: $\text{OT}_d(\cdot, \cdot)$ in §2.1 so that it is only calculated once in the first iteration of Algorithm 1 and reused for subsequent batches. This significantly reduces *SAVA* runtimes with no detriment to performance Figure 9. All experimental results in §5, unless otherwise stated, implement this caching strategy.

**Batch sizes.** If we consider Eq. (7), HOT provides the upper bound for the OT since its optimal coupling is on batches of data points. HOT recovers the OT when either batch only has one support or each dataset only has one batch. Consequently, up to a certain batch size, when increasing the batch size $N_i$, HOT converges to the OT, but its memory complexity also increases, i.e., $\mathcal{O}(N_i \times N'_j) \rightarrow \mathcal{O}(N \times N')$. On the other hand, if the batch size $N_i$ is too small, the number of batches $K_t$ will be large. As a result, the memory complexity will also be high, i.e., $\text{OT}_{\bar{C}}(\bar{\mu}_t, \bar{\mu}_v)$ in Algorithm 1. Thus, the batch size will trade off the memory complexity and the approximation of HOT to standard OT.

## 5 EXPERIMENTS

We aim to the test two following hypotheses: (i) Can *SAVA* scale and overcome the memory complexity issues which hinder *LAVA* while maintaining similar performance? (ii) Can *SAVA* scale to a large real-world noisy dataset with over a *million* data points?

## 5.1 DATASET CORRUPTION DETECTION

We test the scalability of *SAVA* versus *LAVA* (Just et al., 2023) by leveraging the CIFAR10 dataset, introducing a corruption to a percentage of the training data, but keeping the validation set clean. We then assign a value to each data point in the training set. Following Pruthi et al. (2020); Just et al. (2023), we sort the training examples by their values. An effective data valuation method would rank corrupted examples in a prefix of ordered points. We use the fraction of corrupted data recovered by the prefix of size $N/4$ as our detection rate (see an example in Appendix F, Figure 5).

**Setup.** We consider 4 different corruptions (see Appendix E for details) applied to training data following Just et al. (2023): (i) *noisy labels*, (ii) *noisy features*, (iii) *backdoor attacks* and (iv) *poison detections*. All experiments are run on a Tesla K80 Nvidia GPUs with 12GB GPU RAM. Unless otherwise stated, all results reported are a mean $\pm 1$ standard deviation over 5 independent runs.

**Baselines.** Our main baseline is *LAVA*. For *SAVA* and *LAVA* we use features from a pre-trained ResNet18 (He et al., 2016). For *SAVA*, we use a default batch size of $N_i = 1024$ which is its main hyperparameter.[4] We consider *KNN Shapley* (Jia et al., 2019a), the Shapley value measures the marginal improvement in the utility of a data point and uses KNN to approximate the Shapley value. *KNN Shapley* also uses ResNet18 features to calculate Shapley values, we tune $k$ as its performance is sensitive to this hyperparameter. We consider *TracInCP* (Pruthi et al., 2020) which measures the influence of each training point by measuring the difference in the loss at the beginning versus the end of training. We also compare with *Influence Function* (Koh & Liang, 2017) which approximates the effect of holding out a training point on the test error, it uses expensive approximations of the Hessian to calculate the influence of a training point. We also consider *Data-OOB* (Kwon & Zou, 2023) which uses bagging estimators to value data points, data points are embedded using a pre-trained feature extractor, and the bagging estimator is a decision tree (see Appendix G.1 for implementation details). We finally consider a naive OT baseline which obtains values for data points at a batch level, and aggregates values across validation batches; the baseline essentially performs *LAVA* only at a data point level within a batch and averaging values across validation batches. We call this baseline *Batch-wise LAVA*. Although *Batch-wise LAVA* obtains good corruption detection results, it is very sensitive to the batch size (see Appendix H.6).

**Noisy labels.** We corrupt $30\%$ of the labels in the training set by randomly assigning the target a different label. From Figure 2, we can see that *LAVA* has an out-of-memory (OOM) issue when valuing the full training set of 50K points. In contrast, *SAVA*, *Batch-wise LAVA*, *KNN Shapley* and *Data-OOB* can consistently value and detect all corruptions when inspecting 12.5K ordered samples by values. *Influence Functions* matches the performance of *SAVA*, but is very expensive to run on large datasets. *TracInCP* is unable to detect noisy labels better than random selection, similar to the observations in Just et al. (2023). Consequently, when pruning $30\%$ of the data, *SAVA* can consistently improve its accuracy on the test set as the training set size increases (Appendix H.1).

**Noisy features.** We add Gaussian noise to $30\%$ of the training images to simulate feature corruptions that might occur in real datasets. From Figure 2, *LAVA* obtains good performance for moderate dataset sizes, but for training set sizes above 10K, some runs have OOM errors when embedding the entire training set into memory and when calculating the full cost matrix for OT problem. In contrast, *SAVA* can consistently value and detect corruptions. *Batch-wise LAVA* is also able to scale and gets similar performance to *SAVA*. As can *KNN Shapley* and *Data-OOB*, albeit its detection rate is lower than *SAVA*; as we prune $30\%$ of the data for larger and larger dataset sizes performance of *SAVA* outperforms *KNN Shapley* Figure 7. *TracInCP* and *Influence Functions* both struggle to detect noisy data points, both methods were originally shown to detect noisy labels, so we do not expect them to work well beyond noisy label detection.

**Backdoor attacks.** We corrupt $10\%$ of the data with a Trojan square attack (Liu et al., 2018). *SAVA*, *Batch-wise LAVA* and *KNN Shapley* can scale as the dataset size increases while *LAVA* has an OOM error when valuing the largest dataset with 50k points. *TracInCP* and *Influence Functions* both struggle to detect backdoored training points and have long runtimes. *Data-OOB* also struggles to value data points in this scenario, this is in line with recent surveys which show less strong performance when valuing data with noisy features (Jiang et al., 2023).

---

[4]See Appendix H for the study of the sensitivity of the batch size $N_i$ in *SAVA*.

**Poisoning attacks.** We corrupt $10\%$ of the data with a poison frogs attack (Shafahi et al., 2018). We find in Figure 2 that *SAVA* and *Batch-wise LAVA* can scale and maintain a high detection rate. In contrast, *KNN Shapley* struggles to detect corrupted data points after inspecting $25\%$ of the ordered training data. *TracInCP*, *Data-OOB* and *Influence Functions* struggle to detect the corrupted data.

## 5.2 LARGE SCALE VALUATION AND PRUNING

We test our second hypothesis: whether *SAVA* can scale to a large real-world dataset. We consider the web-scrapped dataset Clothing1M (Xiao et al., 2015) where the training set has over 1M images whose labels are noisy and unreliable. However, the validation set has been curated. Clothing1M is a 14-way image classification problem and has been used for previous work on online data selection (Mindermann et al., 2022). We consider *SAVA* and other data pruning methods as baselines to remove low-value training data points before training a ResNet18 classifier.

We compare to *Batch-wise LAVA*, introduced in §5.1. We also compare to *EL2N* (Paul et al., 2021) which values training points using the loss on several partially trained networks to decide which points to remove. We train 10 models for 10 epochs, we perform a cross-validate a sliding window of values to decide which *EL2N* values to keep (see Appendix G.2.2 for details). We also consider supervised prototypes (Sorscher et al., 2022) which prunes image embeddings according to how similar they look to cluster centers after clustering image embeddings (see Appendix G.2.3 for details).

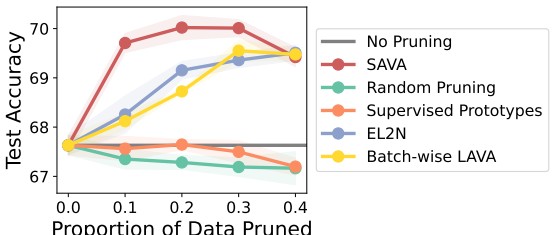

Figure 3: **SAVA can scale to a large web-scrapped dataset.** We use *SAVA* and other baselines, to value data points and then prune a certain percentage of the noisy training set. The resulting dataset is used for training a classifier.

If we were to train a classifier on the full noisy training set, we would obtain an accuracy of $67.6 \pm 0.2$, this remains constant as we randomly prune more data. Supervised prototypes obtains better results than random, and improves slightly over random pruning. This is expected since supervised prototypes is a semantic deduplication method and ignores label information. *SAVA*, *Batch-wise LAVA* and *EL2N* perform well and we find that *SAVA* performs better than both *EL2N* and *Batch-wise LAVA*. This shows the benefit of the SAVA's weighting of the (sub)gradient of the OT across the validation dataset using optimal plan across batches $\bar{\pi}^*(\bar{\mu}_t, \bar{\mu}_v)$ rather than *Batch-wise LAVA*'s uniform weighting. *SAVA* produces the best accuracy model of $70.0 \pm 0.2$ in Figure 3.

## 6 CONCLUSIONS

We have presented a scalable extension to *LAVA* to address the challenges posed by large-scale datasets. Instead of relying on the expensive OT computation on the whole dataset, our proposed *SAVA* algorithm involves jointly optimizing multiple smaller OT tasks across batches of data points and within individual batches. We empirically show that *SAVA* maintains performance in data valuation tasks while successfully scaling up to handle a large real-world noisy dataset. This makes the data valuation task feasible for large-scale datasets. Our proposed method, along with the theoretical correction (Just et al., 2023, Theorem 2), will complement, strengthen and improve (in terms of efficiency) OT-based data valuation.

**Limitations.** HOT finds the optimal coupling at the batch level, but not at the global level as in the traditional OT, i.e., $\text{OT}(\mu_t, \mu_v) \le \text{HOT}(\mu_t, \mu_v)$, which makes the validation error bound looser (Just et al., 2023, Eq. (1), Theorem 1). However, our experiments indicate little performance degradation for small datasets between *SAVA* and *LAVA*. See Appendix B for further discussion on limitations.

## ACKNOWLEDGEMENTS

We thank the area chairs, anonymous reviewers for their comments, and Hoang Anh Just for helpful discussions. TL acknowledges the support of JSPS KAKENHI Grant number 23K11243, and Mitsui Knowledge Industry Co., Ltd. grant.

REPRODUCIBILITY STATEMENT

We have described in detail our algorithm in Section 3 with Algorithm 1. All implementation details for our method and baselines are described in Section 5 and expanded on in Appendix G. The code to reproduce all experiments can be found at https://github.com/skezle/sava.

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

# APPENDIX

In this appendix, we provide discussion regarding the border impact of our work in §A and the limitations of our work in §B. We also provide details of theoretical results in §C, describe the data corruptions used in our experiments in §E. In §F, we further discuss how we calculate detection rates and discuss data valuation rankings. We also give details for the implementations used in the experiments in §G. In §H, we provide further empirical results. In §I, we present and discuss further related works. We also provide a visualization for the artifacts in *SAVA* in §J, and other further discussions in §K.

## APPENDIX A    BROADER IMPACT

Data selection in deep learning for training neural networks can significantly enhance the efficiency and effectiveness of model training. By enabling faster training and improved generalization performance, data selection techniques reduce the computational resources and time required, leading to notable environmental benefits such as lower energy consumption and reduced carbon footprint. By using, an optimal transport approach to data valuation ensures high-quality, relevant data is selected, improving model performance. However, this approach also carries risks: a malicious actor could curate a harmful validation dataset, leading to the training of models on dangerous or unethical data. This underscores the importance of vigilance and ethical considerations in dataset creation and curation.

## APPENDIX B    LIMITATIONS

One theoretical limitation stated in Section 6 regards a looser validation error bound (Just et al., 2023, Eq. (1), Theorem 1) due to the use of hierarchical optimal transport (Yurochkin et al., 2019; Lee et al., 2019). However, the validation error bound for *SAVA* remains useful as in *LAVA* since it can be interpreted that minimizing either the OT or hierarchical OT between training and validation sets, and will bound the model's validation error. As we observe in practice, there is little difference in performance between *SAVA* and *LAVA* (Just et al., 2023). Another limitation of OT-based data valuation methods is their dependence on a clean validation dataset.

Another limitation of our work is that the ground cost we consider is limited to labeled datasets.[5] We have not explored different ground truth costs for text data or speech data, which are interesting directions for future investigation.

## APPENDIX C    DETAILS OF THEORETICAL RESULTS

### C.1    EXISTING THEOREM 2 IN JUST ET AL. (2023)

**Theorem 4** (restated Theorem 2 in Just et al. (2023)). *The difference between calibrated gradients for two data points in $\mathbb{D}_t$ and $\mathbb{D}_v$ can be calculated as*

$$\frac{\partial \operatorname{OT}(\mu_t, \mu_v)}{\partial \mu_t(z_i)} - \frac{\partial \operatorname{OT}(\mu_t, \mu_v)}{\partial \mu_t(z_k)} = \frac{\partial \operatorname{OT}_\varepsilon(\mu_t, \mu_v)}{\partial \mu_t(z_i)} - \frac{\partial \operatorname{OT}_\varepsilon(\mu_t, \mu_v)}{\partial \mu_t(z_k)} - \varepsilon \frac{N}{N-1} \left( \frac{1}{(\pi_\varepsilon^*)_{kj}} - \frac{1}{(\pi_\varepsilon^*)_{ij}} \right) \tag{C.1}$$

The above theorem analyzes the trade-off of using entropic regularization to estimate the OT gradient. However, the theoretical result suffers two key drawbacks that we correct below:

1. The original derivation ignores the positive constraint when deriving its dual formulation: $\pi \geq 0$. This positivity is essential to have the OT plan in the valid domain.

---

[5]For unlabelled datasets, in some sense, it is equivalent to set the weight coefficient $c = 0$, to ignore the contribution of label information for the ground cost.

2. In the proof of (Just et al., 2023, Theorem 2), the authors have incorrectly[6] written the relative entropy (i.e., Kullback-Leibler divergence) term for the discrete case

$$H(\pi_\varepsilon | \mu_t \otimes \mu_v) = \sum_{i=1}^{N} \sum_{j=1}^{N'} \log \frac{(\pi_\varepsilon)_{ij}}{\mu_t(z_i)\mu_v(z_j)}. \tag{C.2}$$

However, the correct term should be written as follows as used in (Cuturi, 2013; Genevay et al., 2016; 2018)

$$H(\pi_\varepsilon | \mu_t \otimes \mu_v) = \sum_{i=1}^{N} \sum_{j=1}^{N'} (\pi_\varepsilon)_{ij} \log \frac{(\pi_\varepsilon)_{ij}}{\mu_t(z_i)\mu_v(z_j)}. \tag{C.3}$$

## C.2 PROOF OF THEOREM 2

**Theorem 5** (restated Theorem 2 in the main paper). *The difference between calibrated gradients for two data points in $\mathbb{D}_t$ and $\mathbb{D}_v$ can be calculated as*

$$\frac{\partial \operatorname{OT}(\mu_t, \mu_v)}{\partial \mu_t(z_i)} - \frac{\partial \operatorname{OT}(\mu_t, \mu_v)}{\partial \mu_t(z_k)} = \frac{\partial \operatorname{OT}_\varepsilon(\mu_t, \mu_v)}{\partial \mu_t(z_i)} - \frac{\partial \operatorname{OT}_\varepsilon(\mu_t, \mu_v)}{\partial \mu_t(z_k)}$$
$$+ \varepsilon \frac{N}{N-1} \left( \log \frac{(\pi_\varepsilon^*)_{ij}/\mu_t(z_i)}{(\pi_\varepsilon^*)_{kj}/\mu_t(z_k)} + \frac{h_{kj}^* - h_{ij}^*}{\varepsilon} \right) \tag{C.4}$$

*where $h^*$ is the corresponding optimal dual variable, accounting for non-negative constraint on the transportation plan $\pi \geq 0$ in the primal OT formulation.*

*Proof.* Using the same notation as in LAVA (Just et al., 2023), the Lagrangian function for the standard OT formulation between datasets $\mathbb{D}_t$ and $\mathbb{D}_v$

$$\mathcal{L}(\pi, h, f, g) = \langle \pi, C \rangle + \langle h, \pi \rangle + \sum_{i=1}^{N} f_i \left( \pi_i^\top \mathbf{I}_{N'} - a_i \right) + \sum_{j=1}^{N'} g_j \left( \mathbf{I}_N^\top \pi_j - b_j \right), \tag{C.5}$$

where we consider the dual variable $h$ for the corresponding constraint $\pi \geq 0$ in the primal OT, which was overlooked in LAVA. Additionally, $f_i$ with $1 \leq i \leq N$ and $g_j$ with $1 \leq j \leq N'$ are corresponding dual variables for the marginal constraints in OT.

The Lagrangian function for entropic regularized OT formulation between datasets $\mathbb{D}_t$ and $\mathbb{D}_v$ is

$$\mathcal{L}_\varepsilon(\pi_\varepsilon, f_\varepsilon, g_\varepsilon) = \langle \pi_\varepsilon, C \rangle + \varepsilon \sum_{i=1}^{N} \sum_{j=1}^{N'} (\pi_\varepsilon)_{ij} \log \frac{(\pi_\varepsilon)_{ij}}{\mu_t(z_i)\mu_v(z_j)}$$
$$+ \sum_{i=1}^{N} (f_\varepsilon)_i \left( (\pi_\varepsilon)_i^\top \mathbf{I}_{N'} - \mu_t(z_i) \right) + \sum_{j=1}^{N'} (g_\varepsilon)_j \left( \mathbf{I}_N^\top (\pi_\varepsilon)_j - \mu_v(z_j) \right), \tag{C.6}$$

where we correct the discrete formulation of entropic regularization (used in LAVA) in the second term on the right-hand side. We have used the following notations:

- $\mathbf{I}_{N'} = (1, 1, \cdots, 1) \in \mathbb{R}^{N' \times 1}$. Similarly, $\mathbf{I}_N \in \mathbb{R}^{N \times 1}$.

- $\pi$ is a primal variable (i.e., the OT assignment matrix or transporation plan). $\pi_i^\top$ is the $i^{\text{th}}$ row of the matrix $\pi$. $\pi_j$ is the $j^{\text{th}}$ column of matrix $\pi$.

- The subscript $\varepsilon$ indicates primal/dual variables for corresponding entropic regularized OT.

Using the first-order necessary condition, we have

$$\nabla \mathcal{L}_\pi(\pi^*, h^*, f^*, g^*) = 0 \tag{C.7}$$
$$\nabla (\mathcal{L}_\varepsilon)_\pi(\pi_\varepsilon^*, f_\varepsilon^*, g_\varepsilon^*) = 0 \tag{C.8}$$

---

[6]This could be a typo within the derivation from Just et al. (2023).

where $\pi^*$ is the optimal solution to the primal formulation, and $(h^*, f^*, g^*)$ are optimal solution to the dual problem.

For $\forall i \in \{1, 2, \ldots, N\}$, and $\forall j \in \{1, 2, \ldots N'\}$, we have

$$\nabla \mathcal{L}_\pi (\pi^*, h^*, f^*, g^*)_{ij} = C_{ij} + h_{ij}^* + f_i^* + g_j^* = 0 \tag{C.9}$$

$$\nabla (\mathcal{L}_\varepsilon)_\pi (\pi_\varepsilon^*, f_\varepsilon^*, g_\varepsilon^*)_{ij} = C_{ij} + \varepsilon \left( \log \frac{(\pi_\varepsilon^*)_{ij}}{\mu_t(z_i)\mu_v(z_j)} + 1 \right) + (f_\varepsilon)_i^* + (g_\varepsilon)_j^* = 0. \tag{C.10}$$

Let's subtract Eq. (C.10) from Eq. (C.9), we have

$$\left[ f_i^* - (f_\varepsilon)_i^* \right] + \left[ g_j^* - (g_\varepsilon)_j^* \right] + h_{ij}^* - \varepsilon \left( \log \frac{(\pi_\varepsilon^*)_{ij}}{\mu_t(z_i)\mu_v(z_j)} + 1 \right) = 0. \tag{C.11}$$

Similarly, $\forall k \neq i, k \in \{1, 2 \ldots N\}$, we have

$$\left[ f_k^* - (f_\varepsilon)_k^* \right] + \left[ g_j^* - (g_\varepsilon)_j^* \right] + h_{kj}^* - \varepsilon \left( \log \frac{(\pi_\varepsilon^*)_{kj}}{\mu_t(z_k)\mu_v(z_j)} + 1 \right) = 0. \tag{C.12}$$

Let's subtract Eq. (C.12) from Eq. (C.11) and rearrange, we have

$$\left[ (f_\varepsilon)_i^* - (f_\varepsilon)_k^* \right] - (f_i^* - f_k^*) = \left( h_{ij}^* - h_{kj}^* \right) - \varepsilon \left[ \log \frac{(\pi_\varepsilon^*)_{ij} / \mu_t(z_i)}{(\pi_\varepsilon^*)_{kj} / \mu_t(z_k)} \right]. \tag{C.13}$$

Additionally, from the definition of the calibrated (sub)gradient, we have

$$\frac{\partial \mathrm{OT} (\mu_t, \mu_v)}{\partial \mu_t(z_i)} - \frac{\partial \mathrm{OT} (\mu_t, \mu_v)}{\partial \mu_t(z_k)} = \frac{N}{N-1} (f_i^* - f_k^*) \tag{C.14}$$

$$\frac{\partial \mathrm{OT}_\varepsilon (\mu_t, \mu_v)}{\partial \mu_t(z_i)} - \frac{\partial \mathrm{OT}_\varepsilon (\mu_t, \mu_v)}{\partial \mu_t(z_k)} = \frac{N}{N-1} \left[ (f_\varepsilon)_i^* - (f_\varepsilon)_k^* \right]. \tag{C.15}$$

Let's substract Eq. (C.14) from Eq. (C.15) and rearrange, we have

$$\frac{\partial \mathrm{OT}_\varepsilon (\mu_t, \mu_v)}{\partial \mu_t(z_i)} - \frac{\partial \mathrm{OT}_\varepsilon (\mu_t, \mu_v)}{\partial \mu_t(z_k)} = \frac{\partial \mathrm{OT} (\mu_t, \mu_v)}{\partial \mu_t(z_i)} - \frac{\partial \mathrm{OT} (\mu_t, \mu_v)}{\partial \mu_t(z_k)} \tag{C.16}$$

$$+ \frac{N}{N-1} \left[ (f_\varepsilon)_i^* - (f_\varepsilon)_k^* - (f_i^* - f_k^*) \right]. \tag{C.17}$$

Plugging Eq. (C.13) in Eq. (C.17), we have

$$\frac{\partial \mathrm{OT}_\varepsilon (\mu_t, \mu_v)}{\partial \mu_t(z_i)} - \frac{\partial \mathrm{OT}_\varepsilon (\mu_t, \mu_v)}{\partial \mu_t(z_k)} = \frac{\partial \mathrm{OT} (\mu_t, \mu_v)}{\partial \mu_t(z_i)} - \frac{\partial \mathrm{OT} (\mu_t, \mu_v)}{\partial \mu_t(z_k)}$$
$$- \varepsilon \frac{N}{N-1} \left( \log \frac{(\pi_\varepsilon^*)_{ij} / \mu_t(z_i)}{(\pi_\varepsilon^*)_{kj} / \mu_t(z_k)} + \frac{h_{kj}^* - h_{ij}^*}{\varepsilon} \right). \tag{C.18}$$

Rearranging it again to be aligned with the result in Just et al. (2023), we conclude the proof

$$\frac{\partial \mathrm{OT} (\mu_t, \mu_v)}{\partial \mu_t(z_i)} - \frac{\partial \mathrm{OT} (\mu_t, \mu_v)}{\partial \mu_t(z_k)}$$
$$= \frac{\partial \mathrm{OT}_\varepsilon (\mu_t, \mu_v)}{\partial \mu_t(z_i)} - \frac{\partial \mathrm{OT}_\varepsilon (\mu_t, \mu_v)}{\partial \mu_t(z_k)} - \varepsilon \frac{N}{N-1} \left( \log \frac{(\pi_\varepsilon^*)_{ij} / \mu_t(z_i)}{(\pi_\varepsilon^*)_{kj} / \mu_t(z_k)} + \frac{h_{kj}^* - h_{ij}^*}{\varepsilon} \right). \tag{C.19}$$

∎

The term $\log \frac{(\pi_\varepsilon^*)_{ij} / \mu_t(z_i)}{(\pi_\varepsilon^*)_{kj} / \mu_t(z_k)}$ is a correction with respect to the corrected discrete formula for entropic regularization. In addition, we have an extra term $\frac{h_{kj}^* - h_{ij}^*}{\varepsilon}$ where $h^*$ is the corresponding optimal dual variable, which accounts for $\pi \geq 0$ in the primal OT formulation.

### C.3 PROOF OF LEMMA 3

We present the Proof of Lemma 3 as follows by building upon the refined theory in Theorem 2 (i.e., the corrected version for Theorem 2 in Just et al. (2023)).

**Lemma 6** (restated Lemma 3 in the main paper). *The difference between the calibrated gradients for two data points $\{z_l, z_h\} \in B_i \subset \mathbb{D}_t$ can be calculated as*

$$\frac{\partial \text{HOT}(\mu_t, \mu_v)}{\partial \mu_t(z_k)} - \frac{\partial \text{HOT}(\mu_t, \mu_v)}{\partial \mu_t(z_l)} = \sum_{j=1}^{K_v} \bar{\pi}_{ij}^*(\bar{\mu}_t, \bar{\mu}_v) \left[ \frac{\partial \text{OT}_\varepsilon(\mu_{B_i}, \mu_{B_j'})}{\partial \mu_{B_i}(z_k)} - \frac{\partial \text{OT}_\varepsilon(\mu_{B_i}, \mu_{B_j'})}{\partial \mu_{B_i}(z_l)} \right.$$
$$\left. + \varepsilon \frac{N_i}{N_i - 1} \left( \log \frac{(\bar{\pi}_\varepsilon^*)_{k,j}/\mu_t(z_k)}{(\bar{\pi}_\varepsilon^*)_{l,j}/\mu_t(z_l)} + \frac{h_{lj}^* - h_{kj}^*}{\varepsilon} \right) \right], \qquad \text{(C.20)}$$

*where $h^*$ is the corresponding optimal dual variable, accounting for nonnegative constraint on the transportation plan (i.e., $\bar{\pi}^* \geq 0$) in the primal OT formulation.*

*Proof.* Let $\text{OT}(\mu_t, \mu_v)$ be the OT formulation between empirical measures $\mu_t$ and $\mu_v$, we present the proof as follows

$$\frac{\partial \text{HOT}(\mu_t, \mu_v)}{\partial \mu_t(z_k)} - \frac{\partial \text{HOT}(\mu_t, \mu_v)}{\partial \mu_t(z_l)}$$
$$= \sum_{j=1}^{K_v} \bar{\pi}_{ij}^*(\bar{\mu}_t, \bar{\mu}_v) \left[ \frac{\partial \text{OT}(\mu_{B_i}, \mu_{B_j'})}{\partial \mu_{B_i}(z_k)} - \frac{\partial \text{OT}(\mu_{B_i}, \mu_{B_j'})}{\partial \mu_{B_i}(z_l)} \right] \qquad \text{(C.21)}$$
$$= \sum_{j=1}^{K_v} \bar{\pi}_{ij}^*(\bar{\mu}_t, \bar{\mu}_v) \left[ \frac{\partial \text{OT}_\varepsilon(\mu_{B_i}, \mu_{B_j'})}{\partial \mu_{B_i}(z_k)} - \frac{\partial \text{OT}_\varepsilon(\mu_{B_i}, \mu_{B_j'})}{\partial \mu_{B_i}(z_l)} \right.$$
$$\left. + \varepsilon \frac{N_i}{N_i - 1} \left( \log \frac{(\bar{\pi}_\varepsilon^*)_{k,j}/\mu_t(z_k)}{(\bar{\pi}_\varepsilon^*)_{l,j}/\mu_t(z_l)} + \frac{h_{lj}^* - h_{kj}^*}{\varepsilon} \right) \right], \text{(C.22)}$$

where Eq. (C.21) follows the definition of HOT and Eq. (C.22) utilizes our Theorem 2. ∎

## APPENDIX D   SAVA TIME COMPLEXITY

Table 2: Time complexity comparison for LAVA and SAVA methods. $V$ and $V'$ are the number of classes in the training and validation sets, the classes are the same for the experiments in this paper. $N$ and $N'$ are the number of points in the training and validation sets, $N_{y_i}$ and $N'_{y_j'}$ are the number of points from training class $y_i$ and validation class $y_j'$. $N_i$ and $N_j'$ are the number of points in the training and validation batch. $K_t$ and $K_v$ are the number of batches in the training and validation sets.

| | Time Complexity |
|---|---|
| *LAVA* | $V \times V' \times N_{y_i} \times N'_{y_j'} \times \log N_{y_i} + N \times N' \times \log N$ |
| *SAVA* | $V \times V' \times N_{i_{y_i}} \times N'_{j_{y_j'}} \times \log N_{i_{y_i}} + K_t \times K_v \times N_i \times N_j' \log N_i + K_t \times K_v \times \log K_t$ |

For simplicity, we assume $N \geq N'$, then Sinkhorn complexity is $\tilde{\mathcal{O}}(N \times N' \times \log N)$ (Dvurechensky et al., 2018).

For *LAVA*, for the training dataset (i.e., big dataset), let $V$ be the number of classes, $N_{y_i}$ be the number of instances in class $y_i$, then the total number of samples $N = \sum_i N_{y_i}$. Note that we use $V', N', N'_{y_j'}$ for the validation dataset (i.e., small dataset), and for simplicity, we assume that $N \geq N'$ and $N_{y_i} \geq N'_{y_j'}$. To compute class-wise Wasserstein distance, we need to solve $V \times V'$ OT problems with the

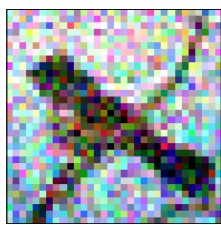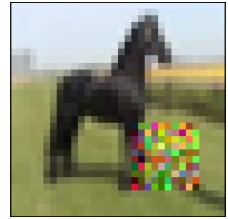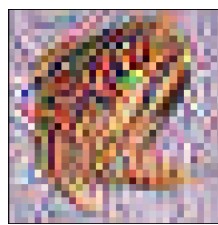

Figure 4: **Examples of the data corruptions used in our experimental setup**. Examples of data from the CIFAR10 dataset where the images have corruptions: noisy features, trojan square, and poison frogs corruptions respectively.

complexity $\tilde{\mathcal{O}}(N_{y_i} \times N'_{y'_j} \times \log N_{y_i})$. Totally, the complexity is $\sum_{i=1}^{V} \sum_{j=1}^{V'} \tilde{\mathcal{O}}(N_{y_i} \times N'_{y'_j} \times \log N_{y_i})$. Additionally, we also need to solve OT between two datasets with complexity $\tilde{\mathcal{O}}(N \times N' \times \log N)$.

For *SAVA*, with the label-to-label caching implementation, for the classwise Wasserstein distance, the total complexity is $\sum_{i=1}^{V} \sum_{j=1}^{V'} \tilde{\mathcal{O}}(N_{iy_i} \times N'_{jy'_j} \times \log N_{iy_i})$ where for each class $y_i, y'_j$, we sample $N_{iy_i}, N'_{jy'_j}$ respectively for these classes. Let $N_i, N'_j$ be the number of samples in batches and $K_t, K_v$ be the number of batches respectively, the total complexity to solve all batch level OT problems is $\sum_{i=1}^{K_v} \sum_{j=1}^{K_t} \tilde{\mathcal{O}}(N_i \times N'_j \times \log N_i)$. Additionally, we need to solve one more OT problem between batches $\tilde{\mathcal{O}}(K_v \times K_t \times \log K_v)$. These time complexities are summarized in Table 2.

**Further reducing computational complexity for Sinkhorn approach.**    One may leverage recent advanced techniques to further speed up the computation of Sinkhorn algorithm, e.g., low-rank approaches (Forrow et al., 2019; Altschuler et al., 2019; Scetbon et al., 2021).

## APPENDIX E    DATA CORRUPTIONS DESCRIPTIONS

We consider 4 different corruptions in our experiments (Section 5) that are applied to the training set following the settings in Just et al. (2023):

**Noisy labels.**    We corrupt a proportion of the labels in the training set by randomly assigning the target a different label. This is a common corruption found in webscale vision (Xiao et al., 2015) and speech (Radford et al., 2023) for instance.

**Noisy features.**    We add Gaussian noise to a certain proportion of the images in the training set to simulate common background noise corruptions that might occur in real datasets.

**Backdoor attacks.**    A certain proportion of the training set is corrupted with a Trojan square attack Liu et al. (2018), e.g., corrupted images have $10 \times 10$ pixel square trigger mask added with random noise and are relabeled to the trojan "airplane" class, see Figure 4 for an example of a corrupted image.

**Poison detections.**    A certain proportion of training data from a specific base class is poisoned such that the features look similar to a target class (Shafahi et al., 2018). Our target is the cat class from the CIFAR10 dataset, which is a randomly chosen image of a cat from the test set. We take a certain percentage of the base class which, in our case is the frog class from the CIFAR10 training dataset. Then we optimize the poisoned images themselves using gradient descent such that the feature spaces are close in Euclidean distance to the target cat image's features using a pre-trained feature extractor model (in our case a pre-trained ResNet18 model). This means that the poisoned images contain features that look like the cat class and the labels are kept the same, meaning that this is a very difficult attack to detect, and the features of frogs are poisoned to look like cats and labels remain uncorrupted. See Figure 4 for an example of a poisoned frog image.

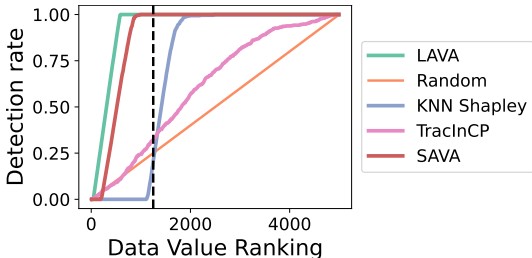

Figure 5: Data value rankings for various methods for the $10\%$ poison frogs corruption. The number of corrupt datapoints in the prefix determines the detection rate. The black dashed line represents the $N/4$ prefix which is used for calculating the detection rates in Figure 2 and Figure 7.

## APPENDIX F  DATA VALUATION RANKINGS

Corrupt data points are likely to be assigned a high value (a low value for KNN Shapley) and so can be used for ranking the data by importance. Therefore, following Pruthi et al. (2020), we sort the training examples by their value in descending order (ascending order for KNN Shapley). An effective data valuation method would rank corrupted examples toward the start of the data valuation ranking. We use the fraction of corrupted data recovered by the prefix of size $N/4$ as our detection rate to measure the effectiveness of various methods in Figure 2. See Figure 5 for an example of this using a poison frogs (Shafahi et al., 2018) corruption on $10\%$ of a dataset of size 5k on CIFAR10.

## APPENDIX G  IMPLEMENTATION DETAILS

### G.1  CIFAR10 CORRUPTION DETECTION

We use a single Nvidia K80 GPU to run all experiments.

**SAVA.**  For both *SAVA* and *LAVA* the metric between label spaces is computed using the the exact OT distances between conditional empirical measures for each label and we do not use Gaussian approximations proposed in Alvarez-Melis & Fusi (2020).

**TracInCP.**  In practice, TracInCP measures the influence of a training point by the dot product of the gradient of the NN model parameters evaluated on the validation set and the gradient of the NN model parameters evaluated on the specific training points, summed throughout training using saved checkpoints of a ResNet18 model trained for 100 epochs and achieves a test accuracy of $83\%$ and training accuracy of $99\%$. We calculate gradients over the entire model and use every second checkpoint to calculate TracInCP values, these design choices result in fewer approximations than the original implementation (Pruthi et al., 2020).

**Influence Functions.**  We use the following repository for obtaining results on influence functions with default parameters as given at `https://github.com/nimarb/pytorch_influence_functions`.

**KNN Shapley.**  The method is very sensitive to $k$ which is a hyperparameter in the kNN algorithm used to approximate the expensive calculation of the Shapley value. We do a grid search over $k \in \{5, 10, 20, 50, 100, 200, 500, 1{,}000, 2{,}000\}$ on a validation set of size $1{,}000$ and training set of size $2{,}000$, where the validation set is taken from the CIFAR10 training set. Our implementation is the same as that used in the LAVA.

**Data-OOB.**  We use the default hyperparameters and use the implementation, given at `https://github.com/ykwon0407/dataoob`.

**Pruning.**  For the pruning experiments we greedily prune $N/4$ of the ranked points with the lowest value; the highest gradient of the OT for *SAVA* and *LAVA*. We then train a ResNet18 with the SGD

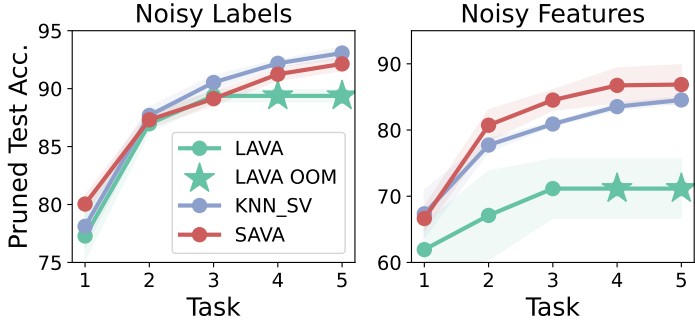

Figure 6: *SAVA can value more data as a dataset incrementally increases in size.* For each task, we value the data and prune $30\%$ of the data which we then train a CNN to obtain a test accuracy. Left: $30\%$ noisy labels setting. Right: $30\%$ noisy feature setting. The star symbol (★) denotes the point at which *LAVA* is unable to continue valuing training due GPU out-of memory errors.

optimizer with weight decay of $5 \times 10^{-4}$ and momentum of $0.9$ for 200 epochs with a learning rate schedule where for the first 100 epochs the learning rate is $0.1$, then for the next 50 epochs the learning rate is $0.01$, then the final 50 epochs the learning rate is $0.001$.

## G.2 CLOTHING1M

For all experiments, we use a node with 8 Nvidia K80 GPUs.

We use a ResNet18 model for feature extraction and for obtaining a final performance metric. We use an Adam optimizer with a weight decay of $0.002$. Since the pruned datasets can be of different sizes depending on the amount of pruning. We train for a fixed number of 100k steps. We use a learning rate schedule where for the first 30k steps the learning rate is $0.1$ then the next 30k steps the learning rate is $0.05$ then the next 20k steps the learning rate is $0.01$, then the next 10k steps the learning rate is $0.001$, then the next 5k steps the learning rate is $0.0001$ then for the final 5k steps the learning rate is $0.00001$.

### G.2.1 SAVA

*SAVA* has remarkably few design choices in comparison to other data pruning methods like EL2N (Appendix G.2.2) and supervised prototypes (Appendix G.2.3). We train a ResNet18 encoder using the clean training set of size $47{,}570$. Note we do not use this training set to obtain final accuracies in Figure 3, we only use the large noisy training set for obtaining the results in Figure 3. We use a batch size of 2048 for valuation and we use label-2-label matrix caching (Appendix H).

### G.2.2 EL2N

To obtain values for the points in our noisy training set to then decide which training points to prune, we obtain our *EL2N* scores by training for 10 epochs 10 separate ResNet18 models on the clean training set of size $47{,}570$.

We also hyperparameter optimize the offset $\in \{0.0, 0.05, 0.1\}$ using a sliding window which covers $90\%$ of points §4 (Paul et al., 2021). This is to decide which range of ranked values to keep/ prune. We find that using an offset of $0.0$ worked best, so high values of the EL2N score will get pruned.

### G.2.3 SUPERVISED PROTOTYPES

We train an image encoder using a classification objective on the clean training set provided in the Clothing1M dataset of size $47{,}570$. Note, that we do not use this dataset to train to augment the pruned noisy training sets after valuation and so are not used for the results in Figure 3.

We use mini-batch k-means clustering to obtain centroids for our image embeddings. We tune the mini-batch k-means learning rate over the grid $\{0.1, 0.05, 0.01, 0.005, 0.001\}$ and the number of

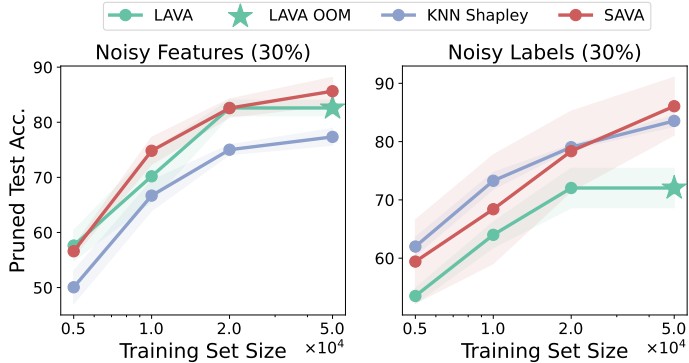

Figure 7: *SAVA can scale to the full CIFAR10 dataset enabling better data selection performance*. After valuation, we prune a prefix of size $N/4$ and train a ResNet18 model on the remaining points to report test accuracy. The symbol (★) denotes the point at which *LAVA* is unable to continue valuing training due to GPU out-of memory (OOM) errors.

---

**Algorithm 2** Incremental learning experimental setup.

---

**Input:** noisy training dataset initally empty $\mathbb{D}_t := \varnothing$ and clean validation set $\mathbb{D}_v$, data pruning proportion $p \in [0, 1]$.
**Output:** trained model $\mathcal{M}$.
**for** $\mathcal{T}_t = 1, ..., T$ **do**
    Get new data $\mathbb{D}_{\mathcal{T}_t}$.
    Merge $\mathbb{D}_{\mathcal{T}_t}$ with existing dataset $\mathbb{D}_t := \mathbb{D}_t \cup \mathbb{D}_{\mathcal{T}_t}$.
    Get valuation scores for $\mathbb{D}_t$ by comparing to $\mathbb{D}_v$.
    Prune a proportion $p$ with the highest data values: $\tilde{\mathbb{D}}_t$.
    Train model $\mathcal{M}$ on $\tilde{\mathbb{D}}_t$ and evaluate $\mathcal{M}$ on $\mathbb{D}_v$.

---

centroids over the grid $\{1,000, 2,000, 5,000, 10,000\}$ using the intra cluster mean-squared error on the validation dataset.

The best configuration uses a k-means clustering learning rate of $0.05$ and $10,000$ cluster centers. Then we can obtain cosine distances between every data point and its assigned cluster center and prune points which look the most prototypical before training a classifier on the pruned dataset.

## APPENDIX H ADDITIONAL EXPERIMENTS

### H.1 CORRUPTION EXPERIMENTS PRUNING PERFORMANCE

We can prune the $N/4$ data points and train a classifier on the pruned dataset in Figure 7. *SAVA* can value a larger and larger pool of training data. The subsequently pruned dataset provides better and better performance as more clean data—which resembles the validation set—is used for training. In contrast, *LAVA* has an OOM issue for the largest dataset when running the Sinkhorn algorithm on a training set of size 50K. As a result, the performance of the ResNet18 model does not improve when valuing larger training sets with *LAVA*.

### H.2 INCREMENTAL LEARNING

We split the CIFAR10 dataset into $5$ equally sized partitions with all classes, and we incrementally build up the dataset such that it grows in size as one would train a production system (Baby et al., 2022). We inject the data with noisy labels and noisy feature corruptions, then perform the data valuation. We compare our proposed method with *LAVA* (Just et al., 2023). After valuing our training set which is incrementally updated and grows in size throughout the incremental learning, we greedily discard $30\%$ of the data that are the most dissimilar to the validation set and train on the pruned

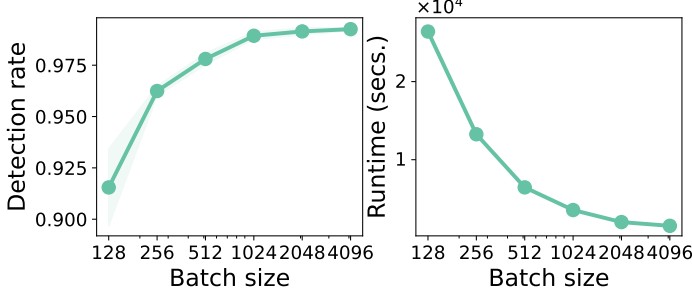

Figure 8: **SAVA performance as function of the batch size,** $N_i$. The CIFAR10 dataset with noisy label detection. Left: Detection rate. Right: detection runtimes in seconds.

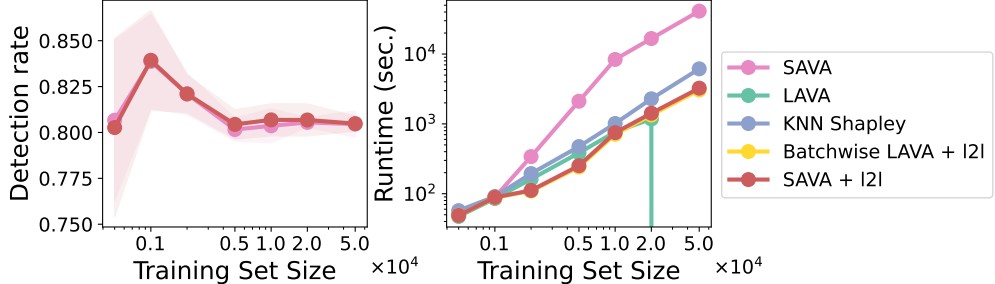

Figure 9: **Label-to-label matrix caching significantly reduces runtime**. Left: *SAVA* with and without label-2-label (l2l) matrix caching performs similarly in detecting noisy label corruption. Right: *SAVA* with l2l runs just a quickly as *LAVA* in terms of runtime in seconds on the same GPU.

dataset. We report the final accuracy of a ResNet18 (He et al., 2016) model. We summarize this experimental setup in Algorithm 2.

## H.3 PERFORMANCE AS A FUNCTION OF BATCH SIZE

Following the discussion on the batch size in Section 4, we measure our model performance w.r.t. different batch sizes $N_i \in \{128, 256, ..., 4,096\}$. We show that the performance converges with increasing batch sizes. In this particular setting, the batch sizes of $1,024, 2,048$, and $4,096$ will perform comparably in terms of detection rate while the batch size of $4,096$ will consume less time for computation of the cost across batches, since there are less and $K_t$ is lower.

## H.4 LABEL-TO-LABEL DISTANCE CACHING

We study the difference in performance and runtime between *SAVA* Algorithm 1 and using *SAVA* with label-to-label (l2l) matrix caching discussed in Section 4 using CIFAR10 with noisy label corruptions. We show that there are almost identical detection rates between the two variants of *SAVA* with and without l2l caching Figure 9. Meanwhile, the runtime is significantly reduced using this caching strategy and is similar to *LAVA* despite having to solve a quadratic number of small batch-level OT problems.

## H.5 CONSTRUCTING BATCHES

We explore two options for constructing the batches including random sampling and stratified sampling in Figure 10. Since (uniformly) random sampling could result in a batch with an uneven distribution of classes, this could degrade the calculation of the class-wise Wasserstein distance Eq. (8). To mitigate, against any imbalance we compare random sampling versus stratified sampling which evenly samples from each class to construct a batch. When valuing 10k points from the CIFAR10 dataset with corrupted features, we find little difference in detection rates between these

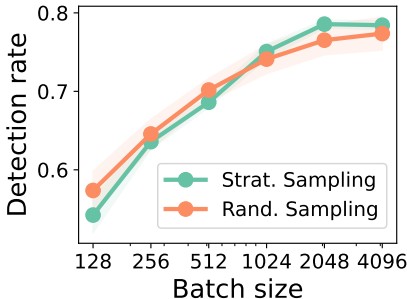

Figure 10: **Randomly sampling data for batch construction is robust**. Detection rates for *SAVA* for random sampling to construct a batch versus stratified sampling which evenly samples data from different classes. The detection rates are calculated after valuation by inspecting the fraction of corrupted data for a prefix of size $N/4$ for CIFAR10 with noisy features.

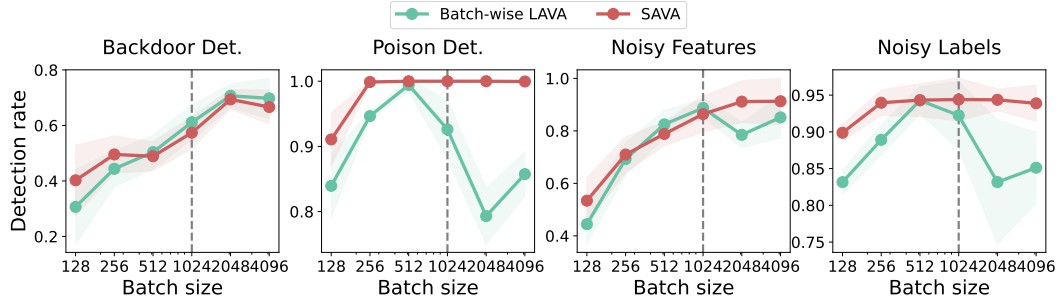

Figure 11: *Batch-wise LAVA* **is not robust to different batchsizes**. For 4 different corruptions on a training dataset of size 5k with validation dataset size 2k we measure the detection rate for varying batch sizes for *SAVA* and *Batch-wise LAVA*. The dashed grey line centered on 1024 denotes the batch size used in Figure 2.

two sampling schemes. This might be a consideration when the ratio of the number of classes in the dataset to the batch size is higher.

## H.6    ON THE ROBUSTNESS OF BATCH-WISE LAVA

From the corruption detection experiments, we established that *Batch-wise LAVA* has remarkable performance in Section 5.1. However, as we vary the batch-size we notice that for small batch-sizes and large batch-sizes *Batch-wise LAVA* performance deteriorates dramatically Figure 11. For small batches this is due to *Batch-wise LAVA* not having enough clean points in the validation batch to detect corrupt data points in the training batch. For large batches the final batch in the dataloader is usually smaller than the specified batch-size and so these data points in the final batch will also suffer from not enough points in the validation batch to compare against. *SAVA* is able to overcome this issue since the information from all batches is aggregated and weighted by the optimal plan between batches, $\pi_{ij}^*(\bar{\mu}_t, \bar{\mu}_v)$, in lines 7 and 10 in Algorithm 1.

## APPENDIX I    RELATED WORKS

**Data valuation.** A simple way to value a datapoint is through the leave-out-out (LOO) error; i.e. the change in performance when the point is omitted, this is inefficient to perform in practice. Influence functions (Koh & Liang, 2017) approximate LOO retraining using expensive approximations of the Hessian of the NN weights. In a similar vein, TracIn (Pruthi et al., 2020) traces the influence of a training point on a test point by looking at the difference in the loss throughout training. Another way to value data is by using data Shapley values Ghorbani & Zou (2019); Jia et al. (2019b), extensions include the Beta Shapley (Kwon & Zou, 2022). K-nearest neighbour models can address the computation cost of the data Shapley (Jia et al., 2019a; Wang et al., 2023; 2024). Instead of using

the Shapley value one can use "the core" from the game theory literature for data valuation (Yan & Procaccia, 2021). An entire dataset can be valued by its diversity, practically this can be done by assessing its volume: the determinant of the left Gram (Xu et al., 2021). An initialized model can also be utilized for data valuation where sets are available from contributors (Wu et al., 2022). The Banzhaf value can also be used for data valuation (Wang & Jia, 2023). Data valuation using out-of-bag estimators have also been shown to be effective (Kwon & Zou, 2023). Our approach builds upon *LAVA* which uses the gradient OT distance between the validation and training sets to assign a value to training points (Just et al., 2023). The OpenDataVal benchmark is available with many implementations of data valuation methods (Jiang et al., 2023).

**Active learning.** Active learning is characterized by selecting points from an unlabeled pool of data for labeling and then using the newly labeled datapoint to update a model (Settles, 2009). Active learning is related to data valuation since a notion of importance is needed to value unlabeled points to then select a label. Unlabeled samples can be valued by using the prediction disagreement from a probabilistic model (Houlsby et al., 2011), this disagreement can also be obtained from multiple models (Freund et al., 1997). This approach of using the disagreement of a probabilistic model can also be thought of as selecting points to label which are the most uncertain via the predictive entropy using probabilistic neural networks (Gal et al., 2017; Kirsch et al., 2019). Alternatively picking points to label can be thought of as a summarization problem by obtaining a coreset of representative data (Sener & Savarese, 2018; Mirzasoleiman et al., 2020; Coleman et al., 2019). Samples to be labeled can be selected by interpreting the classifier output probabilities as a confidence (Li & Sethi, 2006).

**Data selection.** Active learning is used to select points to label and so its importance metric doesn't use label information, however, it has been shown to achieve competitive results for data selection; speeding up the generalization curve over the course of training (Park et al., 2022). The influence a point has on the training loss as a metric of informativeness has been shown to accelerate the training of neural networks (Loshchilov & Hutter, 2015). Similarly picking points that reduce the variance of the gradient speeds up training (Katharopoulos & Fleuret, 2018; Johnson & Guestrin, 2018). Instead of focusing on the training loss, one can select data according to the influence on the validation loss (Mindermann et al., 2022). Instead of selecting data to train on, one can equivalently prune away uninformative data. The data's contribution of the gradient norm of the loss with respect to model parameters is a natural measure for deciding which datapoints to prune from a dataset (Paul et al., 2021). One can also prune data by how similar embeddings are to a cluster center or prototype (Sorscher et al., 2022) and by assessing diversity within each cluster (Abbas et al.; Tirumala et al., 2023). It has also been shown that pruning data according to how easy they are to be forgotten over the course of training - as a measure of difficulty - results in training on less data while maintaining performance (Toneva et al., 2018). These data selection methods although related, do not directly measure the importance of each training datapoint with respect to a clean validation set like *LAVA* (Just et al., 2023) and *SAVA*. Meta-learning is also used to learn datapoint importance weights by evaluating with a clean validation set (Ren et al., 2018). Similarly to *LAVA* and *SAVA* the distributional distance between a clean validation set and a large noisy dataset can be assessed using n-grams in NLP for selecting data to train large language models (Xie et al., 2023).

## APPENDIX J  SAVA VISUALIZATION

To gain insight into how the estimated OT matrices from our proposed *SAVA*, we visualize the artifacts Algorithm 1 in Figure 12.

## APPENDIX K  OTHER DISCUSSIONS

**Overfitting and Approximation.** Recently, Peyré & Cuturi (2019, §8.4) revealed an important property that solving exactly the OT problem may lead to overfitting. Therefore, investing excessive efforts to compute OT exactly would not only be expensive but also self-defeating since it would lead to overfitting within the computation of OT itself. As a result, our batch approximation can be considered as a regularization for OT. We show empirically for certain cases that the batch approximation (*SAVA*) performs better than the original OT (*LAVA*) in terms of quality while we surpass *LAVA* in memory requirement.

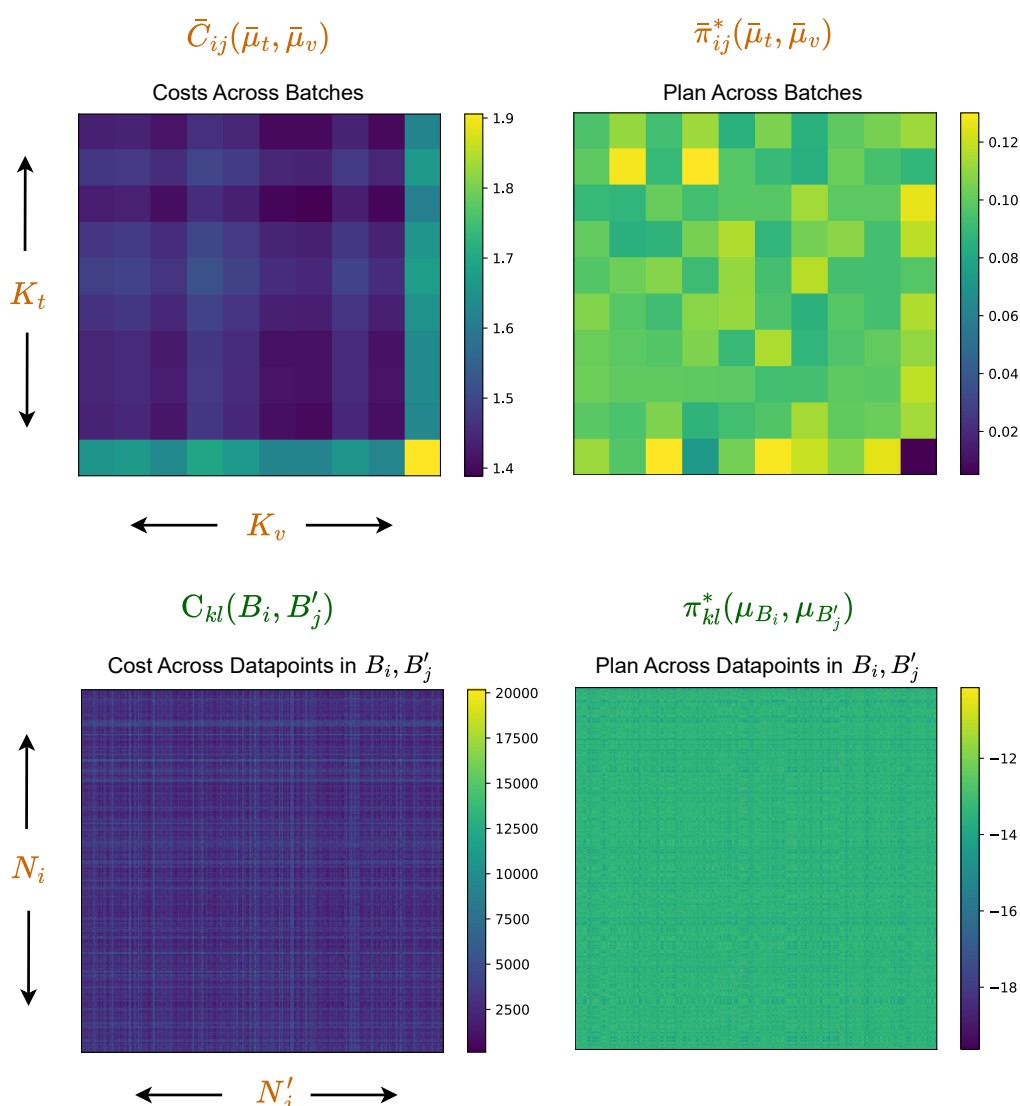

Figure 12: **Visualization of the main artifacts in Algorithm 1**. For a noisy CIFAR10 valuation problem with a training and validation set size of 10k and *SAVA* batch size of 1024, we visualize the main artifacts of the *SAVA* algorithm for illustrative purposes. From left to right, from the top row to the bottom row: the first matrix is the cost matrix between batches: $\bar{C}(\bar{\mu}_t, \bar{\mu}_v)$ and then the optimal plan $\bar{\pi}^*(\bar{\mu}_t, \bar{\mu}_v)$ is the associated plan which is the solution to the optimal transport problem. In the second row we visualize $C(\mu_{B_i}, \mu_{B'_j})$ the optimal transport distance between points in the final batch $B_i$ from the training set and the final batch $B'_j$ in the validation set and its corresponding optimal plan $\pi^*(\mu_{B_i}, \mu_{B'_j})$, we have used a $\log$ transform to on the optimal plan between datapoints to help viewing it.

