# OpenReview forum: "SAVA: Scalable Learning-Agnostic Data Valuation"
_ICLR.cc/2025/Conference — ICLR 2025 Poster_

### Official Review · Reviewer_RqVV · 2024-10-30

**Soundness:** 3
**Presentation:** 3
**Contribution:** 3
**Rating:** 6
**Confidence:** 4

**Summary:**

This paper proposes a new learning-agnostic data valuation approach that assigns a value to each data point in the training set based on its similarity to the validation set. They introduce SAVA, a scalable variant of the LAVA algorithm, which uses optimal transport (OT) sensitivity to value training data without directly concerning model performance efficiently. Unlike LAVA requiring the entire dataset as input, SAVA operates on batches of data points, making it has a smaller memory consumption. Thus, SAVA can make the valuation taks of larger datasets possible. The authors conduct extensive experiments showing that SAVA can effectively scale to large datasets and maintain data valuation performance across various downstream tasks.

**Strengths:**

[S1] An interesting approach leveraging the idea of batches to solve the memory bottleneck encountered in OT solver as optimizer in model training.

[S2] Detailed theoretical proofs and descriptions of previous work are given.

[S3] The article is well-organized and easy to read.

**Weaknesses:**

[W1] My biggest concern is the proof of the upper bound does not adequately explain why this proxy can work.  Detailed analysis on the upper bound of the proxy practicability should be taken.

[W2] My second concern is that the paper lacks of time complexity analysis. And SAVA in Figure 2 seems to be no better than Batch-wise LAVA. In the appendix Figure 9, why not compare Batch-wise LAVA in running time metric?

[W3] Typos: Line 417, "Batch-wise LAVA KNN Shapley and" -> "Batch-wise LAVA, KNN Shapley, and"

Placing Table 1 in Section 2 would help to improve understanding.

**Questions:**

pls see W2

---

> ### Author Response · Authors · 2024-11-21
> **Response to review (1/n)**
>
> **Q: [W1] My biggest concern is the proof of the upper bound does not adequately explain why this proxy can work.  Detailed analysis on the upper bound of the proxy practicability should be taken.**
>
> A: Apologies for the confusion, but we are not entirely clear on what is meant by “the proof of the upper bound does not adequately explain why this proxy can work”. For instance:
> * Which "upper bound" are you referring to here?
> * What is the “proxy” you are referring to in your question?
> Many thanks in advance.
>
> Intuitively, our approach by using (sub)gradient of HOT to value input data points is theoretically supported by a direct result from Theorem 1 (Just et al., 2023) and the fact that OT is upper-bounded by HOT (line 259). More concretely, Theorem 1 (Just et al., 2023) says that the discrepancy between the training and validation performance of a model is bounded by OT between the training and the validation datasets. Therefore, this discrepancy between the training and validation performance of a model is also bounded by the HOT between the training and the validation datasets since OT is in fact upper-bounded by HOT (line 259).
>
> Additionally, the purpose of our Theorem 2 (i.e., the correction of Theorem 2 (Just et al., 2023)) for LAVA and corresponding Lemma 3 for SAVA is to characterize the exact trade-off for using entropic-regularization to approximate the gradient of OT/HOT respectively.
>
> **Q: [W2] My second concern is that the paper lacks of time complexity analysis.**
>
> A: Many thanks for the suggestion, we have added a comparison of the time complexity of SAVA versus LAVA in section D of the appendix.
>
> * For simplicity, we assume $N \ge N'$, then Sinkhorn time complexity is $O(N \times N’ \times \log(N))$ [Dvurechensky 2018].
> * Using the notation from the paper:
>    * $N$ is the size of the training set.
>    * $N’$ the size of the validation set.
>    * $V$ is the number of classes.
>    * $K_t$ the number of training batches.
>    * $K_v$ the number of validation batches.
>    * $N_{y_i}$ the number of instances per class $y_i$ such that $N = \sum_i N_{y_i}$.
>    * $N_{i_{y_i}}$  the number of instances per class $y_i$ per batch.
>    * $\mathbf{V}, \mathbf{N}$, $\mathbf{N_\mathbf{y_j}}$, $\mathbf{N_{i_{y_i}}}$ is used to denote the respective validation dataset quantities. In the paper we use the prime notation $V’, N', N'_{y'_j}$, but in OpenReview we use the bold notation due to Latex rendering issues.
>
> For LAVA, we compute class-wise Wasserstein distance and solve $V \times V'$ OT problems with the complexity $\tilde{\mathcal{O}}(N_{y_i} \times \mathbf{N_\mathbf{y_j}} \times \log N_{y_i}$). Totally, the complexity is $\sum_{i=1}^{V} \sum_{j=1}^{\mathbf{V}} \tilde{\mathcal{O}}(N_{y_i} \times\mathbf{N_\mathbf{y_j}} \times \log N_{y_i})$. Additionally, we also need to solve OT between two datasets with complexity $\tilde{\mathcal{O}}(N \times \mathbf{N} \times \log N)$.
>
> For SAVA, with the label-to-label caching implementation, for the classwise Wasserstein distance, the total complexity is $\sum_{i=1}^V \sum_{j=1}^{\mathbf{V}} \tilde{\mathcal{O}}(N_{i_{y_i}} \times \mathbf{N_{i_{y_i}}} \times N_{i_{y_i}})$ where for each class $y_i, \mathbf{y_j}$, we sample $N_{i_{y_i}}, \mathbf{N_{i_{y_i}}}$ respectively for these classes. Let $N_{i}, \mathbf{N_{j}}$ be the number of samples in batches and $K_t, K_v$ be the number of batches respectively, the total complexity to solve all batch level OT problems is $\sum_{i=1}^{K_v} \sum_{j=1}^{K_t} \tilde{\mathcal{O}}(N_i \times \mathbf{N_{j}} \times \log N_i)$. Additionally, we need to solve one more OT problem between batches $\tilde{\mathcal{O}}(K_v \times K_t \times \log K_v)$.
>
> [Dvurechensky 2018] Dvurechensky, Pavel, Alexander Gasnikov, and Alexey Kroshnin. "Computational optimal transport: Complexity by accelerated gradient descent is better than by Sinkhorn’s algorithm." International conference on machine learning. PMLR, 2018.
>
>
> **Q: [W2] And SAVA in Figure 2 seems to be no better than Batch-wise LAVA.**
>
> A: Indeed, we find it remarkable that it performs so well for the corrupted CIFAR10 experiments, it is an interesting result for the OT community, potentially these CIFAR10 experiments are too easy. That said, batch-wise LAVA is more sensitive to the batch size hyper-parameter, Fig 11, especially for small batch sizes due to small numbers of points in the validation batch to compare to, and with large batch sizes since the final batch might be small too. Crucially, batch-wise LAVA underperforms SAVA for the large scale Clothing1M experiment.

---

> > ### Author Response · Authors · 2024-11-21
> > **Response to review (2/n) n =2**
> >
> > **Q: [W2] In the appendix Figure 9, why not compare Batch-wise LAVA in running time metric?**
> >
> > We have updated Figure 9 to include Batch-wise LAVA in the run-time analysis. We have updated all methods to ensure we benchmark on the same hardware. Our implementation of Batch-wise LAVA also uses label-to-label caching and so is as performant as SAVA with label-to-label caching although SAVA requires solving a final OT problem in line 6 of Alg 1. Using label-to-label caching is essential to reduce runtimes by an order of magnitude for SAVA and matching LAVA runtimes.
> >
> > **Q: [W3] Typos: Line 417, "Batch-wise LAVA KNN Shapley and" -> "Batch-wise LAVA, KNN Shapley, and"**
> >
> > A: Many thanks for pointing this out, we have corrected this typo.
> >
> > **Q: Placing Table 1 in Section 2 would help to improve understanding.**
> >
> > A: We have placed Table 1 in the main body of the paper to improve readability.

---

> > > ### Author Response · Authors · 2024-11-24
> > > **Any Questions from the Reviewer RqVV on our Rebuttal?**
> > >
> > > We would like to thank the Reviewer again for your thoughtful comments and valuable feedback.
> > >
> > > We would appreciate it if you could let us know whether our responses have addressed your concerns, and whether you still have any other questions about our rebuttal.
> > >
> > > We would be happy to do any follow-up discussion or address any your additional comments.

---

> > > > ### Author Response · Authors · 2024-11-25
> > > > **Kind Reminder: Response of Author Rebuttal for Paper 6869**
> > > >
> > > > Dear Reviewer RqVV,
> > > >
> > > > We sincerely appreciate the time you have taken to provide feedback on our work, which has helped us greatly improve its clarity, among other attributes.
> > > >
> > > > This is a gentle reminder that the discussion phase will end in less than 2.5 days from this comment, i.e., 11:59 pm AoE on November 26. We are happy to answer any further questions you may have before then. Please note that you cannot respond to us after that time, and we cannot reply to you after 11:59 pm AoE on November 27.
> > > >
> > > > If our responses have addressed your concerns, we kindly ask that you consider raising your score to reflect your updated evaluation of our paper more accurately. Thank you again for your time and thoughtful comments!
> > > >
> > > > Sincerely,
> > > >
> > > > The Authors

---

> > > > > ### Author Response · Authors · 2024-11-30
> > > > > **Final Reminder: Response of Author Rebuttal for Paper 6869**
> > > > >
> > > > > Dear Reviewer RqVV ,
> > > > >
> > > > > We appreciate the time you have taken to provide feedback on our work.
> > > > >
> > > > > This is a final reminder that the discussion phase will end soon, on Monday 2 December. We would be more than happy to answer any further questions you may have before then.
> > > > >
> > > > > If our responses have addressed your concerns, we kindly ask that you consider revising your score to reflect these clarifications. Thank you again for your time and thoughtful comments!
> > > > >
> > > > > Sincerely,
> > > > >
> > > > > The Authors of Paper 6869

---

### Official Review · Reviewer_ckNn · 2024-11-03

**Soundness:** 3
**Presentation:** 3
**Contribution:** 2
**Rating:** 6
**Confidence:** 3

**Summary:**

This paper develops a variant of LAVA, called SAVA, for scalable data valuation. The idea is perform data valuation on batches of data instead of on the entire dataset. Extensive numerical results are presented to demonstrate SAVA's efficiency.

**Strengths:**

- The experimental results are convincing. The authors compared to SOTA methods for data valuation across various data corruption scenarios. The results demonstrate that SAVA is scalable to large datasets. Also, the results included a dataset of size larger than 1 million samples, in which the proposed method outperforms benchmarks.

- The writing is good and easy to follow.

**Weaknesses:**

- The reviewer's biggest concern is related to novelty. Currently, SAVA seems a very natural extension of LAVA for data valuation on batches. The submission seems to be on the incremental side, unless the authors can clearly state the technical challenge when calculating on batches.

- The choice of batch size is a key hyper-parameter in SAVA (and key difference to LAVA). The authors are suggested to include formal theoretical analysis to quantify the tradeoff in choosing batch size between memory and calculation approximation. Also, Appendix G should appear in the main text.

- The authors are suggested to include a table comparing the complexities of LAVA and SAVA.

- What happens if the validation dataset gets corrupted?

- In Fig. 3, why is the performance of SAVA dropping at .4 proportion?

**Questions:**

See weaknesses.

---

> ### Author Response · Authors · 2024-11-21
> **Response to review (1/n)**
>
> **Q: The reviewer's biggest concern is related to novelty. Currently, SAVA seems a very natural extension of LAVA for data valuation on batches. The submission seems to be on the incremental side, unless the authors can clearly state the technical challenge when calculating on batches.**
>
> A: The main technical challenges are:
> 1. We derive the (sub)gradient of the hierarchical OT for scalable data valuation using batch-level OT solutions, Lemma 1.
> 2. We correct and refine Theorem 2 from Just et al. 2023 to illustrate the trade-off when one uses entropic regularized OT to approximate standard OT w.r.t. the (sub)gradient in our proposed SAVA method (in Lemma 3).
> 3. Empirically  we demonstrate that SAVA can value data sets of $O(10^6)$ with our experiments using Clothing1M. To our knowledge, this is the largest dataset considered in the data valuation literature. SAVA can identify noisy data points and prune them to subsequently train a model to obtain the best test accuracy (while it is prohibited for the seminal LAVA (Just et al. 2023)).
>
> **Q: The choice of batch size is a key hyper-parameter in SAVA (and key difference to LAVA). The authors are suggested to include formal theoretical analysis to quantify the tradeoff in choosing batch size between memory and calculation approximation. Also, Appendix G should appear in the main text.**
>
> A: We thank the Reviewer for the suggestions. To our knowledge, there is no formal theoretical analysis of the batch-size in the HOT literature yet. This is still an open research question that touches on domains beyond data valuation. For the moment, we suggest a practitioner should perform hyper-parameter optimization for the batch size on a subset of the dataset to obtain the desired performance. This was our approach.
>
> **Q: The authors are suggested to include a table comparing the complexities of LAVA and SAVA.**
>
> A: Thank you for the suggestion, we have included a table in a revised version of our manuscript with the following time complexities in Section D of the appendix.
>
> * For simplicity, we assume $N \ge N'$, then Sinkhorn time complexity is $O(N \times N’ \times \log(N))$ [Dvurechensky 2018].
> * Using the notation from the paper:
>    * $N$ is the size of the training set.
>    * $N’$ the size of the validation set.
>    * $V$ is the number of classes.
>    * $K_t$ the number of training batches.
>    * $K_v$ the number of validation batches.
>    * $N_{y_i}$ the number of instances per class $y_i$ such that $N = \sum_i N_{y_i}$.
>    * $N_{i_{y_i}}$  the number of instances per class $y_i$ per batch.
>    * $\mathbf{V}, \mathbf{N}$, $\mathbf{N_\mathbf{y_j}}$, $\mathbf{N_{i_{y_i}}}$ is used to denote the respective validation dataset quantities. In the paper we use the prime notation $V’, N', N'_{y'_j}$, but in OpenReview we use the bold notation due to Latex rendering issues.
>
> For LAVA, we compute class-wise Wasserstein distance and solve $V \times V'$ OT problems with the complexity $\tilde{\mathcal{O}}(N_{y_i} \times \mathbf{N_\mathbf{y_j}} \times \log N_{y_i}$). Totally, the complexity is $\sum_{i=1}^{V} \sum_{j=1}^{\mathbf{V}} \tilde{\mathcal{O}}(N_{y_i} \times\mathbf{N_\mathbf{y_j}} \times \log N_{y_i})$. Additionally, we also need to solve OT between two datasets with complexity $\tilde{\mathcal{O}}(N \times \mathbf{N} \times \log N)$.
>
> For SAVA, with the label-to-label caching implementation, for the classwise Wasserstein distance, the total complexity is $\sum_{i=1}^V \sum_{j=1}^{\mathbf{V}} \tilde{\mathcal{O}}(N_{i_{y_i}} \times \mathbf{N_{i_{y_i}}} \times N_{i_{y_i}})$ where for each class $y_i, \mathbf{y_j}$, we sample $N_{i_{y_i}}, \mathbf{N_{i_{y_i}}}$ respectively for these classes. Let $N_{i}, \mathbf{N_{j}}$ be the number of samples in batches and $K_t, K_v$ be the number of batches respectively, the total complexity to solve all batch level OT problems is $\sum_{i=1}^{K_v} \sum_{j=1}^{K_t} \tilde{\mathcal{O}}(N_i \times \mathbf{N_{j}} \times \log N_i)$. Additionally, we need to solve one more OT problem between batches $\tilde{\mathcal{O}}(K_v \times K_t \times \log K_v)$.
>
> [Dvurechensky 2018] Dvurechensky, Pavel, Alexander Gasnikov, and Alexey Kroshnin. "Computational optimal transport: Complexity by accelerated gradient descent is better than by Sinkhorn’s algorithm." International conference on machine learning. PMLR, 2018.

---

> > ### Author Response · Authors · 2024-11-21
> > **Response to review (2/n) n=2**
> >
> > **Q: What happens if the validation dataset gets corrupted?**
> >
> > A: SAVA (and LAVA, Just et al. 2023) require a clean validation dataset to value data points. If the validation dataset is corrupted then SAVA will assign a high value to corrupted data in the training set. This is indeed a limitation of our method which we acknowledge in Appendix B. Note that we only need a **small** clean validation dataset to value all data points in a **large** noisy training dataset (illustrated in Fig. 1).
> >
> > **Q: In Fig. 3, why is the performance of SAVA dropping at .4 proportion?**
> >
> > A: We observe an increase in performance when comparing test performance after training on a pruned dataset (10%-40% of the data) versus training on the entire dataset (0% pruning). The optimal pruning percentage seems to be around 20%-30%. For more aggressive pruning, 40%, we start to see performance decrease (but still better than 0% pruning, no pruning) as data that is not noisy is starting to be removed from the dataset. At some pruning percentage we can expect performance starting to deteriorate so it is not surprising to see performance dropping after a certain amount of pruning. Since this is a large non-curated dataset, it is difficult to gauge the exact percentage of truly noisy data points to gauge a “correct” pruning percentage.

---

> > > ### Author Response · Authors · 2024-11-24
> > > **Any Questions from the Reviewer ckNn on Our Rebuttal?**
> > >
> > > We would like to thank the Reviewer again for your thoughtful comments and valuable feedback.
> > >
> > > We would appreciate it if you could let us know whether our responses have addressed your concerns, and whether you still have any other questions about our rebuttal.
> > >
> > > We would be happy to do any follow-up discussion or address any your additional comments.

---

> > > > ### Author Response · Authors · 2024-11-25
> > > > **Kind Reminder: Response of Author Rebuttal for Paper 6869**
> > > >
> > > > Dear Reviewer ckNn,
> > > >
> > > > We sincerely appreciate the time you have taken to provide feedback on our work, which has helped us greatly improve its clarity, among other attributes.
> > > >
> > > > This is a gentle reminder that the discussion phase will end in less than 2.5 days from this comment, i.e., 11:59 pm AoE on November 26. We are happy to answer any further questions you may have before then. Please note that you cannot respond to us after that time, and we cannot reply to you after 11:59 pm AoE on November 27.
> > > >
> > > > If our responses have addressed your concerns, we kindly ask that you consider raising your score to reflect your updated evaluation of our paper more accurately. Thank you again for your time and thoughtful comments!
> > > >
> > > > Sincerely,
> > > >
> > > > The Authors

---

> > > ### Comment · Reviewer_ckNn · 2024-11-27
> > > **Response**
> > >
> > > The reviewer appreciates the authors' efforts for this response. While most of my concerns/comments are addressed, I am still concerned about the novelty. Why is calculating the sub-gradient of OT a challenge? Wouldn't this be a direct extension from LAVA?
> > >
> > > Given that said, however, I am willing to lift my score to 6 to acknowledge the improvement from the initially submitted version.

---

> ### Author Response · Authors · 2024-11-28
> **Thanks for your endorsement!**
>
> Thank you for your response, and we deeply appreciate your thoughtful endorsement.
>
> Our answer for your questions is as follow:
>
> + We agree with the Reviewer that our approach based on hierarchical OT is an extension of LAVA, but respectfully, it is not trivial. Indeed, note that **a naive HOT approach is not entirely efficient since its runtime to compute HOT subgradient is not necessarily faster than the LAVA approach**. Our proposed label-to-label cost caching significantly reduces the runtime of the naive HOT approach with no detriment to performance (see line 401-406 and Figure 9). Briefly, **the proposed label-to-label cost caching is simple, but critical to make our approach based on HOT effective and efficient for data valuation**, especially for large-scale settings (where it is prohibited for the seminal LAVA).
>
> + Theoretically, following OT theory, we derive the HOT subgradient (to our knowledge, it has not been done in previous HOT approaches in the OT literature). Importantly, we **correct and refine the trade-off** by using entropic regularization to approximate the computation of OT subgradient in LAVA to characterize the corresponding exact trade-off for our approach based on HOT.
>
> If our responses have addressed your concerns, we kindly ask that you consider raising your score to reflect your updated evaluation of our paper more accurately. Again, thank you very much for your time and thoughtful comments!

---

### Official Review · Reviewer_94Rm · 2024-11-04

**Soundness:** 3
**Presentation:** 3
**Contribution:** 3
**Rating:** 8
**Confidence:** 4

**Summary:**

This paper investigates of the problem of extending Optimal Transport (OT) distance-based data valuation methods for larger scale problems. The paper points out that for current methods, the quadratic overhead for expensive GPU memory constrained the scale of problems they can be applied to. Correspondingly, this paper proposes to compute the OT problem in a batch-wise fashion where the batch-wise results are aggregated via an hierarchical OT framework to produce data point valuations. This approach allows converting intractable large-scale OT problems into a series of smaller problems that can be solved efficiently. Empirical results on a variety of tasks show the proposed approach achieves competitive performance compared to original methods while being applicable to larger-scale problems.

**Strengths:**

The problem is well-contextualized and the motivation is clear. Structure of the paper is well balanced and the elaborations are coherent. It is straightforward for readers to understand the scope and target of the paper and the proposed technical approaches.

The proposed method is plausible, leveraging the hierarhical OT framework to aggregate results from batch-wise OT computations and achieving favorable approximation results.

Derivations are comprehensive and are paired with substantial elaborations. Empirical evaluations are diverse and abundant and the results are valid.

**Weaknesses:**

I am still somewhat concerned about the computation overhead for SAVA. Even it avoids directly solving large-scale OT problems and circumvents OOM issues, it now requires solving a quadratic number of OT problems between every pair of batches and aggregating their results. This could also take a significant amount of time if the number of batches are high.

Are there results on actual time comparisons for the methods in empirical studies?

The structure of the paper still has room to improve. The current layout is dense where there are many equations and lemmas interleaved with elaborations. There's an overhead for the readers to familiarize with the notations before being able to catch up with the ideas. It could be made more straightforward.

For example, the crucial Figure 1 and Algorithm 1 are not self-contained. Many of the involved notations are not straightforward and also not introduced in the captions. It still requires readers to first read through the texts and derivations to understand what is being done. Strongly suggests authors to make an effort to improve these visualizations, which could substantially improve the paper's accessibility and impact.

**Questions:**

Other than hierarchical OT and the proposed implementation, there are some other ideas for mitigating OT efficiency issues.

Some standard approaches include low-rank approximation to the transportation matrix C, which is often possible for practical cases. This allows representing the large matrix C with multiplication of smaller matrices and avoids directly materilizing the large matrix C and OOM issues.

Another somewhat connected idea is to directly quantize the train and validation distributions (e.g., approximate the distributions via downsampling) to simplify the OT problem.

Hierarchical OT can also be conducted with clustering methods. For example, at the lower level, group all the samples into a number of clusters, and at the higher level, solve the OT problem between the centroids of clusters.

It will be very interesting to see how to connect the proposed framework to these ideas and whether they may help further improving the computation complexity or accuracy.

---

> ### Author Response · Authors · 2024-11-21
> **Response to review (1/n)**
>
> **Q: I am still somewhat concerned about the computation overhead for SAVA. Even it avoids directly solving large-scale OT problems and circumvents OOM issues, it now requires solving a quadratic number of OT problems between every pair of batches and aggregating their results. This could also take a significant amount of time if the number of batches are high.**
>
> **Are there results on actual time comparisons for the methods in empirical studies?**
>
>
> A: Indeed SAVA allows us to perform data valuation on large datasets by solving OT problems on individual batches allowing us to work with datasets which were previously inaccessible to OT-based data valuation methods (LAVA). Since we need to solve OT problems at batch-level, this means that we have to solve $K_t \times K_v$ smaller OT problems where $K_t$ and $K_v$ are the number train and validation batches. Indeed this can add an overhead in terms of runtimes for data valuation.
>
> We provide a runtime comparison of SAVA versus LAVA in Fig 9. We have updated this figure to include batchwise LAVA as requested by reviewer RqVV and re-run all experiments on the same hardware for equal comparison.
>
> If we compare runtimes of LAVA and SAVA then SAVA takes a long time to run. It turns out that we can be more efficient by caching the class level cost matrix (the 2nd term in Eq 7) and sharing the class level cost between all batch-level OT problems, this results in the significant speed up with runtimes matching LAVA (denoted SAVA + l2l also introduced in lines 401-406). Thus the runtime for solving a large OT problem in LAVA is similar to solving many smaller batch-level OT problems with label-to-label cost matrix caching. All results in the paper use this label-to-label caching strategy.
>
> We note that solving batch-level OT problems is also parallelizable. Our implementation does not parallelize the batch-level OT problems (lines 1-5 Alg 1) of SAVA but this is an extension that can be used in future work for further runtime efficiency.
>
> **Q: The structure of the paper still has room to improve. The current layout is dense where there are many equations and lemmas interleaved with elaborations. There's an overhead for the readers to familiarize with the notations before being able to catch up with the ideas. It could be made more straightforward.**
>
> **For example, the crucial Figure 1 and Algorithm 1 are not self-contained. Many of the involved notations are not straightforward and also not introduced in the captions. [...].**
>
> A: We thank the reviewer for the suggestion on improving the readability of the paper. We have added a caption for Alg 1 and improved the caption in Fig 1 in the text. We have moved Table 1 into the main text to aid the reader to familiarize themselves with the notation.
>
> **Q: Other than hierarchical OT and the proposed implementation, there are some other ideas for mitigating OT efficiency issues.**
>
> A: We discuss this in lines 207-212. For instance, sliced-Wasserstein (Rabin et al., 2011) or Sobolev transport (Le et al., 2022) are good approaches for approximating the OT distance, but for data valuation, we require the (sub)gradient of the OT w.r.t. to a mass of input support. We also need to solve the dual formulation of the OT for the optimal dual variables, which may not be obtainable for sliced-Wasserstein (while Sobolev transport limits for input support data points on a given graph structure). That is why we opt for hierarchical OT.

---

> ### Author Response · Authors · 2024-11-21
> **Response to review (2/n) n=2**
>
> **Q: Some standard approaches include low-rank approximation to the transportation matrix C, which is often possible for practical cases. This allows representing the large matrix C with multiplication of smaller matrices and avoids directly materilizing the large matrix C and OOM issues.**
>
>
> A: We agree that the low-rank approach for OT is an alternative direction for making LAVA more memory efficient [1, 2, 3]. Moreover, low-rank approximations can also complement our HOT framework, e.g., we can use a low-rank approach to approximate OT between batches and scale up the HOT approach further.
>
> Recall that we need the (sub)gradient of the OT for data valuation. Therefore, a trade-off for using the low-rank approach for Sinkhorn is that the OT approximation may become looser than using the traditional Sinkhorn, this could have a detrimental impact on data valuation performance. This is an interesting research direction that we leave for future investigation.
>
>
> [1] Forrow, Aden, Jan-Christian Hütter, Mor Nitzan, Philippe Rigollet, Geoffrey Schiebinger, and Jonathan Weed. "Statistical optimal transport via factored couplings." In The 22nd International Conference on Artificial Intelligence and Statistics, pp. 2454-2465. PMLR, 2019.
>
> [2] Altschuler, J., Bach, F., Rudi, A. and Niles-Weed, J., 2019. Massively scalable Sinkhorn distances via the Nyström method. Advances in neural information processing systems, 32.
>
> [3] Scetbon, Meyer, Marco Cuturi, and Gabriel Peyré. "Low-rank sinkhorn factorization." In International Conference on Machine Learning, pp. 9344-9354. PMLR, 2021.
>
> **Q: Another somewhat connected idea is to directly quantize the train and validation distributions (e.g., approximate the distributions via downsampling) to simplify the OT problem.**
>
> A: We did not try to quantize the datasets, however, the OT problems are solved using NN embedding features, not on the datasets themselves. We did investigate lowering the precision of the features. Current OT solvers all use float32 precision inputs, we investigated whether we could use mixed-precision inputs (float16) together with current OT solvers. Unfortunately, current solvers do not support lower precision data types (to the best of our knowledge). However, lowering the precision is not quite the same as quantizing input features. The trade-off in lowering the memory requirement through quantization with the potential lower performance is left for future investigation.
>
> **Q: Hierarchical OT can also be conducted with clustering methods. For example, at the lower level, group all the samples into a number of clusters, and at the higher level, solve the OT problem between the centroids of clusters.**
>
> A: In case one uses clustering to quantize input labelled data points into clusters, then consider datasets as measures over clusters (via the cluster centroids), and only compute the OT between those measures over clusters for data valuation. Consequently, one can only value the clusters (or more precisely, the cluster centroids), but not the input labelled data points anymore.
>
> Additionally, we agree that clustering is an alternative approach to partition input labelled data points into batches instead of using the random partitions for HOT, but it comes with the extra cost of clustering. Empirically, we observe that HOT works well with random partitions into batches. It would be interesting to reconcile clustering into the proposed LAVA and SAVA frameworks and see whether it helps to further improve the performances, with the extra cost from clustering to partition labelled data points into batches. Therefore, we leave this trade-off on using clustering for HOT for future investigation.
>
> We further note that for some clustering methods, e.g., K-means clustering method, we would be required to compute the means for **labelled** data points, which is nontrivial and may be very costly. More precisely, we need to compute the mean for labelled data points $(x_i, y_i)$ w.r.t. the label-feature distance in Eq. (1).

---

> ### Author Response · Authors · 2024-11-24
> **Any Questions from the Reviewer 94Rm on Our Rebuttal?**
>
> We would like to thank the Reviewer again for your thoughtful comments and valuable feedback.
>
> We would appreciate it if you could let us know whether our responses have addressed your concerns, and whether you still have any other questions about our rebuttal.
>
> We would be happy to do any follow-up discussion or address any your additional comments.

---

> > ### Author Response · Authors · 2024-11-25
> > **Kind Reminder: Response of Author Rebuttal for Paper 6869**
> >
> > Dear Reviewer 94Rm,
> >
> > We sincerely appreciate the time you have taken to provide feedback on our work, which has helped us greatly improve its clarity, among other attributes.
> >
> > This is a gentle reminder that the discussion phase will end in less than 2.5 days from this comment, i.e., 11:59 pm AoE on November 26. We are happy to answer any further questions you may have before then. Please note that you cannot respond to us after that time, and we cannot reply to you after 11:59 pm AoE on November 27.
> >
> > If our responses have addressed your concerns, we kindly ask that you consider raising your score to reflect your updated evaluation of our paper more accurately. Thank you again for your time and thoughtful comments!
> >
> > Sincerely,
> >
> > The Authors

---

> ### Comment · Reviewer_94Rm · 2024-11-30
>
> I thank the authors for the responses. I have carefully read through the rebuttals, revisions in the manuscript, and other reviewers & responses. I remain positive about this work and recommend it for publishing. Specifically, the discussions in this rebuttal are very insightful and informative. I recommend the authors add them to the paper or its appendix.
>
> Based on the scores and other reviews, I believe this paper is headed to being accepted. **I wish to give it a score of 7, Clear Accept, but this option isn't available.** The reason I'm hesitant about giving it a score of 8 is I feel its presentation can still be improved. The current paper is technically solid with a valid contribution, but I'm concerned whether it could effectively reach out to a broader audience beyond the data valuation community. I hope the authors can continue to improve it for maximum impact.
>
> **I raised my score to 8. Should the AC hesitate about whether to accept the paper for its borderline scores, I'm willing to advocate for its acceptance.**

---

> ### Author Response · Authors · 2024-11-30
> **Thank you for your endorsement!**
>
> Many thanks for your response, and we deeply appreciate your thoughtful endorsement.
>
> We will revise the paper following the feedback and suggestion in the rebuttal discussion.
>
> With best regards,

---

### Author Response · Authors · 2024-11-21
**Global response to reviews**

We express our gratitude to the Chairs and the Reviewers for spending time reviewing our paper and providing constructive feedback. We are grateful to the Reviewers for recognizing firstly the importance of the problem setting and the novelty of SAVA which scales the OT data valuation problem by using batches (94Rm, RqVV), secondly the clarity of the presentation (ckNn, RqVV), thirdly the comprehensive derivations (94Rm, RqVV), and finally the empirical analysis (94Rm, ckNn). We are also grateful for constructive critical comments, which helped us to improve the paper!

# Paper updates

Many thanks for your suggestions. We have made the following revisions to our paper and updated the paper on OpenReview highlighting our changes in orange.

## Clarity of the derivations (94Rm, RqVV)

We have revised the captions for Fig 1 and Alg 1 to make them clearer and self-contained. We have also moved the notation table from the appendix into the main body of the paper to make it easier for the reader to familiarize themselves with the notation.

## Time complexity (ckNn, RqVV)

We have added a new Section D in the appendix where we compare the time complexity of SAVA and LAVA.

## Runtimes (94Rm, RqVV)

Given the suggestion from RqVV to include batch-wise LAVA to the runtime analysis in Fig 9. We have updated the plot to benchmark runtimes on the same GPU, we have updated the curves for SAVA, LAVA, KNN Shapley and SAVA with label-to-label caching (used throughout the paper). We have added the new runtime curves for batch-wise LAVA with label-to-label caching (used throughout the paper). Although there is a quadratic number of small batch-level OT problems which need to be solved, runtimes are comparable to SAVA when using label-to-label caching.

---

### Meta-Review · Area_Chair_EtJZ · 2024-12-20

**Metareview:**

This paper introduces SAVA, a new and more scalable variant of the data valuation method LAVA. LAVA was restricted by the amount of computation required when computing the optimal transport distance. SAVA addresses this problem by computing these metrics in a batch manner. Experiment results successfully showed the promises of the approach.

**Additional Comments On Reviewer Discussion:**

Reviewers are unanimous about the paper's contribution.

---

### Decision · Program_Chairs · 2025-01-22

Accept (Poster)